# Metabolic capabilities mute positive response to direct and indirect impacts of warming throughout the soil profile

Nicholas C. Dove [1,2]✉, Margaret S. Torn [3], Stephen C. Hart [4] & Neslihan Taş [3,5]✉

Increasing global temperatures are predicted to stimulate soil microbial respiration. The direct and indirect impacts of warming on soil microbes, nevertheless, remain unclear. This is particularly true for understudied subsoil microbes. Here, we show that 4.5 years of whole-profile soil warming in a temperate mixed forest results in altered microbial community composition and metabolism in surface soils, partly due to carbon limitation. However, microbial communities in the subsoil responded differently to warming than in the surface. Throughout the soil profile—but to a greater extent in the subsoil—physiologic and genomic measurements show that phylogenetically different microbes could utilize complex organic compounds, dampening the effect of altered resource availability induced by warming. We find subsoil microbes had 20% lower carbon use efficiencies and 47% lower growth rates compared to surface soils, which constrain microbial communities. Collectively, our results show that unlike in surface soils, elevated microbial respiration in subsoils may continue without microbial community change in the near-term.

[1] Environmental Systems Graduate Group, University of California, Merced, CA, USA. [2] Biosciences Division, Oak Ridge National Laboratory, Oak Ridge, TN, USA. [3] Earth and Environmental Sciences Area, Lawrence Berkeley National Laboratory, Berkeley, CA, USA. [4] Department of Life & Environmental Sciences and Sierra Nevada Research Institute, University of California, Merced, CA, USA. [5] Biosciences Area, Lawrence Berkeley National Laboratory, Berkeley, CA, USA. ✉email: ndove7@gmail.com; ntas@lbl.gov

The impact of increasing global temperatures on soil microbial communities and carbon dioxide ($CO_2$) emissions is still a great source of uncertainity[1], and this uncertainty is exacerbated in the subsoil. Deep soil microbial communities are largely understudied compared to their surface counterparts[2]. Most soil warming studies focus on the upper soil layers where microbial activity and carbon (C) concentrations are significantly higher[3,4]. However, the microbial response to soil warming at depth is non-negligible. For instance, over half of extracellular enzyme activity in the upper meter occurs below 20 cm[4]. Furthermore, Hicks Pries et al.[5] recently showed that the apparent $Q_{10}$ (i.e., temperature sensitivity) of the microbial community was similar throughout the soil profile. When warmed, subsoil respiration accounted for 40% of the increase in $CO_2$ emissions from the whole-soil profile[5]. Additionally, microbial community structure and composition in the subsoil are different than those in the surface soils, as subsoil microbial communities are dominated by slow-growing oligotrophs[6,7]. Under a warming climate, uncertainties surrounding the trajectory of subsoil microbial community reorganization limit our predictive capability of future microbial states and functionalities.

The indirect effects of increased temperatures on soil microbial communities and microbial respiration further complicates our understanding of climate-carbon cycle feedbacks, making it difficult to constrain long-term models of soil-C cycling[8]. For example, while short-term measurements indicate that increased temperatures enhance microbial respiration, long-term (>10y) in situ measurements of microbial respiration are more nuanced[9,10]. Besides the direct effects of increased temperature, in situ soil warming can also decrease C availability[11,12], alter organic matter composition[13], and increase nutrient availability[11] in surface soils. These impacts are likely to affect rates of respiration through indirect effects on the soil microbial community. For instance, decreased C quantity and quality (i.e., number of enzymatic steps necessary to depolymerize C compounds) alters microbial community composition and decreases respiration[14,15]. Additionally, increased nutrient availability in warmed soils decreases extracellular enzyme production for nutrient-acquiring enzymes[16], which may increase biomass growth and also alter microbial community structure. Indeed, long-term soil warming has been shown to alter surface soil microbial community structure and metabolism[17,18], but it is unclear to what extent this is caused by increased temperatures or in combination with altered availability of resources for microbial growth such as organic substrates and nutrients[19]. Understanding changes to microbial metabolism in response to soil warming may, in part, resolve some of these uncertainties, constrain model predictions[8], and explain the disparate findings of long-term empirical observations[1,9,10].

Warming-induced changes to resource availability—due to increased decomposition and altered plant growth—could have an exacerbated effect on the subsoil microbial community where resource demand is the strongest. Resource availability at depth differs from surface soil, where C and nitrogen (N) availabilities decrease with depth[3] because of smaller pool sizes, increased mineral protection, and increased spatial discontinuity of organic matter[20]. However, laboratory rates of C and N mineralization of added substrates were as fast in the subsoils as in surface soils in an old-growth forest, suggesting that microbial competition and demand for C and N resources does not decrease with depth[21]. Hence, subsoil microbial communities may have as strong of substrate demands as in the surface, but mineralization in subsoil is more substrate-limited. Because over half of soil organic C (SOC) is found below a 20 cm depth[3], incorporating subsoil responses to warming is critical in constraining predictions for long-term soil-C storage.

Another important parameter for models of soil-C storage is C use efficiency (CUE). Microbial CUE describes the partitioning of metabolized C that is used for growth (possibly leading to soil-C storage) or respiration (leading to soil-C loss)[15]. While many long-term soil-C models parameterize microbial CUE as constant over time and across space[1], the validity of this assumption is unknown. For instance, differences in resource availability lead to altered CUE by virtue of the C cost associated with extracellular enzyme production. Furthermore, CUE is partially genetically constrained[22], so shifts in microbial community structure with environmental change (i.e., increased temperatures, altered resource availabilities) or with depth may also impact microbial CUE. Yet, our understanding for how CUE changes with the cumulative effects of environmental change and with depth is still lacking due to conflicting measurements among studies[23]. Measuring the impacts of increased temperatures and altered resource availability on CUE factorially across depths can help in resolving these disparate results.

In order to determine the direct (i.e., enzyme and growth kinetics) and indirect (i.e., nutrient mediated) impacts of soil warming on microbial community composition, physiology, and metabolism throughout the soil profile, we analyze samples taken from 0 to 80 cm in depth at the Blodgett Experimental Forest in the central Sierra Nevada, CA[5]. Samples were collected from an ongoing in situ deep mineral-soil warming experiment (+4 °C above ambient) after 4.5 years of warming (Fig. 1A). Recent work at this site showed that deep soil warming has enhanced $CO_2$ emissions by 29% throughout the soil profile, resulting in a third less SOC in the subsoil (<20 cm) after 4.5 years (Soong et al. in review). This warming-induced reduction in SOC was mainly attributed to decreased free particulate organic matter (Soong et al. in review), which is more directly available to microbial decomposition[24], highlighting the changing resource availabilities at this site due to warming.

Here, we use DNA sequencing (16S rRNA and ITS genes, shotgun metagenomics), reconstruction of microbial genomes (metagenome-assembled genomes [MAGs]), differential analysis of estimated in situ MAG growth rates, and CUE measurements

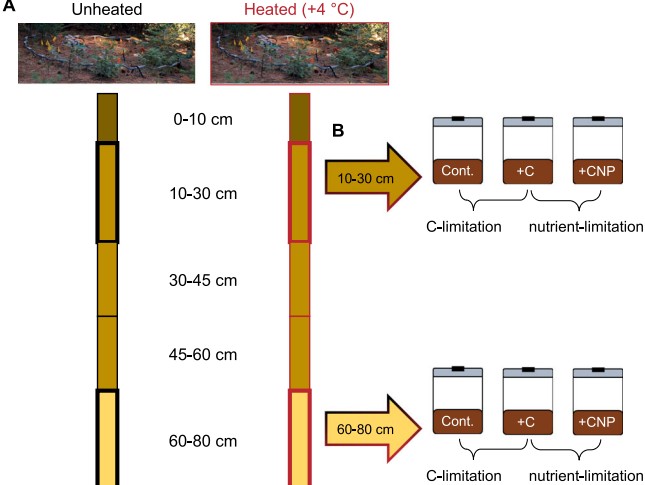

**Fig. 1 Field and laboratory experimental design.** To test for the effect of warming in the field (**A**), we sampled five soil horizons (0–10, 10–30, 30–45, 45–60, and 60–80 cm) from unheated and heated (+4 °C) plots (n = 3). The 10–30 and 60–80 cm horizons from each plot were incubated for 30-d with no amendment (Cont.), cellobiose (+C), or cellobiose with inorganic nitrogen and phosphorus (+CNP) to assess the relative resource limitation of each soil (**B**). The response ratio of +C/Cont. and +CNP/+C was used to determine the relative C- and nutrient limitation, respectively.

via metabolic flux analysis to determine how microbial community structure and metabolism were altered by warming. We couple these analyses with a series of process measurements from laboratory incubations with different C and nutrient amendments to investigate relative resource limitations (Fig. 1B). We hypothesize that: (1) warming-induced soil-C depletion results in increased C limitation and decreased nutrient limitation for the microbial communities in the whole-soil profile; (2) resource limitations shift microbial community composition, physiology, and metabolism towards more oligotrophic traits (e.g., higher microbial CUE, increase in genes encoding for enzyme degrading complex C compounds); and (3) these changes to the microbial community are strongest in the subsoils where resource demand is greatest. We find that microbial communities in heated soils shifted moderately in their composition and metabolism in the upper 30 cm, and were largely C limited compared to unheated soils. Yet, contrary to our hypothesis, subsoil microbial communities responded differently to the direct and indirect effects of soil warming than did surficial microbes—the effect of warming and altered resource availability on microbial community composition and metabolism was relatively weaker in the subsoil. Our results show that elevated subsoil microbial respiration may continue without microbial community turnover and adaptation in the near term. If these conditions persist, delayed temperature acclimation at depth could boost $CO_2$ emissions from subsoil horizons.

## Results

**Resource availability is a key factor affecting surface soil microbial respiration under warming.** We evaluated the impact of altered resource availability on microbial respiration by incubating heated and unheated soils with C and nutrient (N and phosphorus [P]) amendments for 30 days (Fig. 1B). Field-moist soils from each plot and horizon were split into subsamples and given either a control amendment (distilled water), a C amendment, or a C+ nutrient amendment in a 0.4 ml solution in concentrations corresponding to ~10 times the microbial C and N and ~20 times the microbial P found in these soils (Supplementary Table 1). Both in heated and control treatments, soil horizons were tested in triplicate for each amendment resulting in 36 incubations (2 field treatment × 2 soil horizons × 3 amendments × 3 replicates). Using the response ratio of respiration from C-amended to that from unamended soils and the ratio of C- and nutrient-amended (CNP) soils to C-amended soils, we measured the proximate C and nutrient limitation, respectively, of microbial respiration. The "proximate" resource limitation is defined as resource that stimulates an ecosystem process (e.g., microbial respiration)[25]. These ratios measure the effect of added C to quantify proximate C limitation and added nutrients in the context of nonlimiting C to quantify proximate nutrient limitation. We assessed proximate nutrient limitation via the ratio of CNP- to C-amended soils rather than the ratio of nutrient-only amended soils to control soils because added C enhances nutritional demands and, thus, increases the sensitivity in detecting nutrient limitations (see Sullivan et al.[26]). It was not our goal to quantify the ultimate resource limitation. Instead, these ratios are used to determine changes in proximate C and nutrient limitations individually.

We found that microbial respiration in heated surface soils was mostly constrained by C limitation, while unheated surface soils were relatively more nutrient limited. These differences in limitations were only apparent in surface soils (10–30 cm) and not in subsoils (60–80 cm). Cumulative respiration was highest in the CNP-amended soils across depths and heating treatments (Mixed-effects model [MEM]: $p < 0.050$, Fig. 2A). Furthermore,

cumulative respiration was higher in heated surface soils compared to unheated surface soils (MEM: $p < 0.001$), but cumulative respiration was similar between heating treatments in the subsoils (MEM: $p = 0.824$, Fig. 2A). Cumulative respiration was significantly more nutrient limited in unheated surface soils than in heated surface soils (MEM: $p = 0.036$, Fig. 2B), but this heating treatment difference was not apparent in the subsoils (MEM: $p = 0.957$). In contrast, the C limitation of cumulative respiration was not significantly different between heating treatments (MEM: $p = 0.357$) and depths (MEM: $p = 0.120$). Because resource limitations may be most apparent during maximal rates of activity[27], we also analyzed resource limitations of respiration during days 3 and 8, when the rate of respiration was near its maximum (Supplementary Fig. 1). During this early period, there was a day × depth × heating interaction for both C and nutrient limitations (both MEMs: $p < 0.001$). Using multiple comparisons testing (Tukey's test of Honest Significant Differences on the MEM), we found that, during the 8th day of incubation, C limitation was 81% higher in heated surface soils than the unheated counterparts ($p = 0.033$, Fig. 2C). Also, during the 8th day of incubation, unheated surface soils had a 265% higher nutrient limitation than heated surface soils ($p < 0.001$, Fig. 2C). Differences in resource limitations between heating treatments across all other date and depth combinations were not significant ($p > 0.050$), suggesting that differences in the relative resource limitations were only apparent during maximal activity and in surface soils.

**Microbial carbon use efficiency declines with depth but did not significantly respond to warming or nutrient amendment treatments.** We measured microbial CUE at the end of the 30-d resource-amendment incubation using the isotopologue-metabolic modelling approach[28]. We found that across all resource amendment and heating treatments, CUE was 20% lower in the subsoils (range = 0.12–0.59) than in the surface soils (0.39–0.63, MEM: $p < 0.001$; Fig. 2D). This supports predictions of declining CUE with decreased C concentrations due to the greater C cost of resource acquisition at depth[29]. In contrast, altered resource availability (MEM: $p = 0.582$, Fig. 2D) and the heating treatment (MEM: $p = 0.879$) did not change CUE.

**The impact of warming on microbial taxonomic composition is depth dependent and resistant to shifts in resource availability.** Four and a half years of continuous +4 °C warming resulted in modest but significant shifts in bacterial and archaeal (herein referred to as prokaryote; PerMANOVA: $p = 0.005$, $R^2 = 0.076$, Fig. 3A) and fungal (PerMANOVA: $p < 0.001$, $R^2 = 0.094$, Fig. 3B) community composition along the soil-depth profile. While total C, total N, and gravimetric water content (GWC) together correlated with prokaryote community composition, explaining 41% of the variation (distance-based redundancy analysis [db-RDA]: $p = 0.011$), these edaphic variables did not correlate with fungal community composition (db-RDA: $p = 0.667$). However, when assessed individually, none of these edaphic variables correlated with prokaryote community composition (marginal db-RDA: $p > 0.050$).

Responses of the surface soil (0–30 cm) and subsoil (>30 cm) microbial communities were different. For example, in the surface soil, the heating treatment significantly changed prokaryote communities at the operational taxonomic unit (OTU) level, resulting in increases and decreases in OTU abundance within phyla (Fig. 3C). However, in the subsoil, the heating treatment resulted in mostly decreased abundances of multiple OTUs (Fig. 3C), particularly within Bacteroidetes, Gemmatimonadetes, and Verrucomicrobia. For fungi at the phylum level, we only

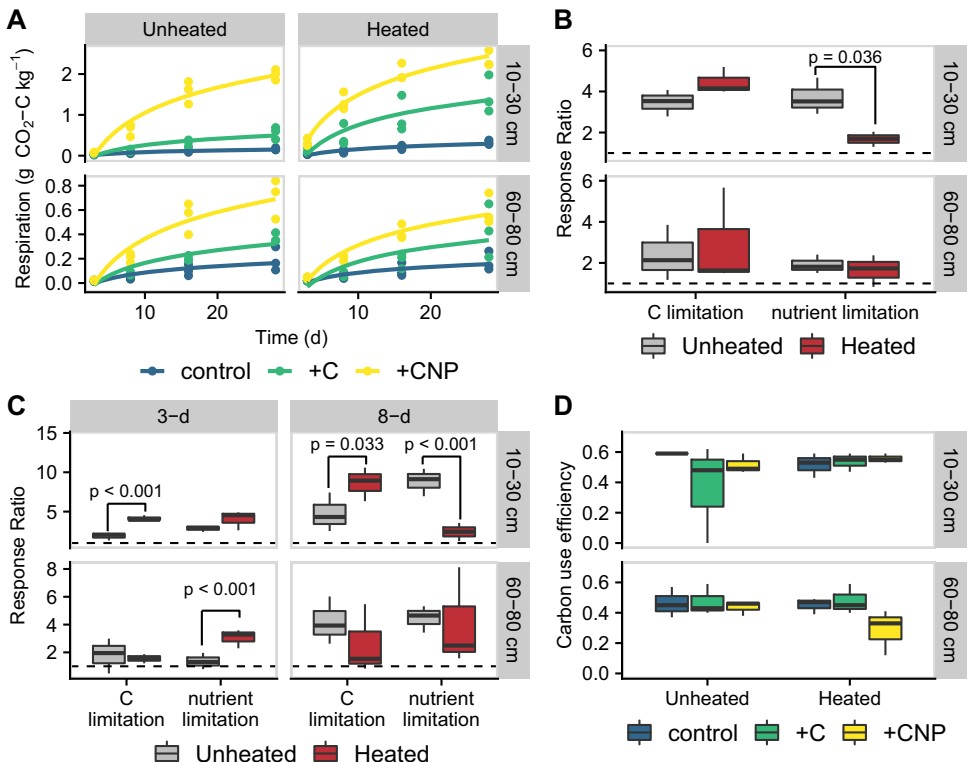

**Fig. 2 Process rate measurements from the incubation experiment. A** Cumulative microbial respiration (with best-fit regression) measured over the 30-d incubation for each amended soil. **B** Boxplots ($n = 3$) show resource limitations of cumulative microbial respiration across both depths and heating treatments. **C** Boxplots ($n = 3$) show resource limitations of microbial respiration at 3 and 8 days of the incubation across both depths and heating treatments. Dashed line at 1.0 in panels **B**, **C** indicates no limitation. **D** Means (and standard error [$n = 3$]) of carbon use efficiency for each amended soil. Carbon limitation (C limitation) is assessed by the response of the carbon-amended soils (+C) divided by the control soil, and nutrient limitation is assessed by the response of the carbon- and nutrient-amended soil (+CNP) divided by the carbon-amended soil (+C). For boxplots (panels **B–D**), the median is indicated by the thick black line, and the lower and upper hinges correspond to the first and third quartiles. The lower and upper whiskers extend to smallest or largest value, respectively, no further than 1.5 times the interquartile range. Asterisk (*) indicates significant differences ($p < 0.05$) between heating treatments (mixed-effects model).

detected a significant change in the ratio of Ascomycota to Basidiomycota abundances in the 0–10 cm depth interval (MEM: $p < 0.001$, Fig. 2D). At the family level, we detected a significant increase in the relative abundance of the ectomycorrhizal (EM) family Tuberaceae throughout the soil profile (MEM: $p = 0.003$, Supplementary Fig. 2). However, the overall impact of heating on EM relative abundance interacted with depth (MEM: $p = 0.029$), such that heating reduced EM relative abundance (largely due to decreased *Inocybe* spp.) only in soils 0–30 cm (MEM: $p = 0.011$, Supplementary Fig. 3). In the subsoils, EM relative abundance was maintained in heated plots due to the significant proportions of the EM families Cortinariaceae and Rhizopogonaceae. Hence, while the relative abundance of EM fungi did not decrease with depth (MEM: $p = 0.711$, Supplementary Fig. 3), the community composition of EM fungi differed with soil depth (PerMANOVA: $p < 0.001$, $R^2 = 0.052$, Supplementary Fig. 2). Sequencing the ITS region, we did not detect arbuscular mycorrhizal fungi in our soils. These results highlight the lack of fungal taxonomic and functional response to warming at depth. Across depths, Blodgett Forest soils were dominated by Acidobacteria, Proteobacteria, and Basidiomycota, but the heating treatment increased the relative abundance of Actinobacteria and Planctomycetes and decreased the relative abundance of Acidobacteria throughout the soil profile (Supplementary Fig. 4). Nevertheless, overall alpha diversity (Chao1 richness and Shannon's diversity) was unchanged by the heating treatment (prokaryote Chao1 MEM: $p = 0.655$, prokaryote Shannon MEM: $p = 0.337$, fungal Chao1

richness MEM: $p = 0.383$, fungal Shannon MEM: $p = 0.159$; Supplementary Fig. 5). We also used a read-based alpha diversity metric (Nonpareil, $D_{NP}$[30]) to compare diversity estimates between amplicon- and metagenome-based sequencing efforts. $D_{NP}$ for all soil depths was $20.7 \pm 0.6$ (standard error of the mean, $n = 30$), falling within the proposed spectrum for soil metagenomes[31]. The $D_{NP}$ was unaltered by soil heating (MEM: $p = 0.904$, Supplementary Table 2). Collectively, these results showed that deep soil warming had no significant impact on microbial richness.

Depth was a significant moderator of prokaryotic (PerMANOVA: $p < 0.001$, $R^2 = 0.191$, Fig. 3A) and fungal (PerMANOVA: $p = 0.004$, $R^2 = 0.054$, Fig. 3B) community composition. At the phylum level, this manifested as an increase in the relative abundance of Chloroflexi (MEM: $p = 0.003$) and a decrease in the relative abundance of Bacteroidetes (MEM: $p = 0.012$, Supplementary Fig. 4). For fungi, the ratio of Ascomycota to Basidiomycota read abundances did not change with depth (MEM: $p = 0.805$, Fig. 3D). While richness decreased significantly with depth for fungi (MEM: $p = 0.002$) and remained constant for prokaryotes (MEM: $p = 0.214$), Shannon's Diversity remained constant for fungi (MEM: $p = 0.227$) and decreased with depth for prokaryotes (MEM: $p = 0.028$, Supplementary Fig. 5). The discrepancies between these two alpha diversity metrics suggest that evenness increased and decreased with depth for prokaryotes and fungi, respectively. The $D_{NP}$ also decreased significantly with soil depth (MEM: $p < 0.001$, Supplementary Table 2).

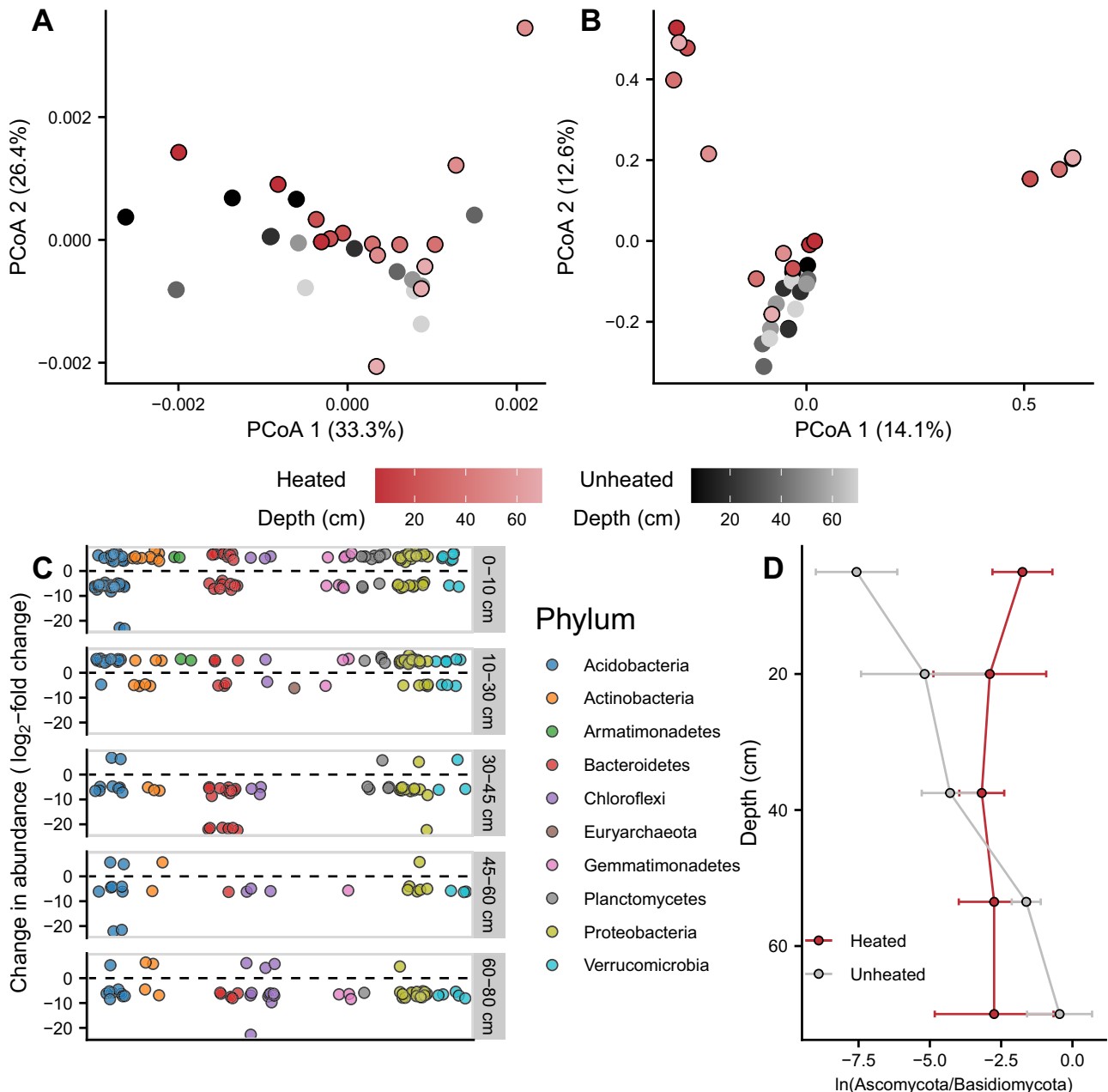

**Fig. 3 Microbial communities change with heating and depth.** Ordinations of microbial communities using Principal Coordinates Analysis (PCoA) for prokaryotes (**A**) and fungi (**B**). Points denote individual samples. Red colors show heated plots and black colors show unheated control plots. Saturated colors show upper depths and unsaturated colors show deeper depths. Prokaryote operational taxonomic units with significant changes in abundance by soil heating (Wald test: $p < 0.05$) are shown by phylum and across depths (**C**). Relative abundances of fungal phyla (i.e., Ascomycota to Basidiomycota ratios) are shown for heated (red) and unheated (gray) soils (**D**). Error bars show ±one standard error ($n = 3$).

In laboratory incubations, the prokaryote but not fungal community composition was modestly altered by the main effect of the resource amendments (prokaryotes PerMANOVA: $p = 0.039$, $R^2 = 0.180$; fungi PerMANOVA: $p = 0.563$, $R^2 = 0.050$; Fig. 4). Changes in prokaryote community composition were most pronounced in the CNP amendments across depths and heating treatments, with increases in the relative abundance of Proteobacteria, Bacteriodetes, and Actinobacteria, and decreases in the relative abundance of Verrucomicrobia and Planctomycetes (Supplementary Fig. 6). In order to create a parallel comparison to respiration responses, we determined a response ratio for the prokaryote community shifts by calculating the Bray-Curtis distance between the beginning and end of the incubation. We did not find a significant difference in C or nutrient limitation between the heating treatments when averaged across both depth zones (i.e., the main effect of the heating treatment; C limitation MEM: $p = 0.757$, nutrient limitation MEM: $p = 0.354$, Supplementary Fig. 7). However, for each depth × heating treatment × plot combination, there was, on average, a greater nutrient than C limitation in the prokaryote community composition shift (paired $t$-test: $p < 0.001$). Across depth and heating treatments, the average C response ratio was 0.97 (SE = 0.06, 1 = no effect) while the average nutrient response ratio was 1.40 (SE = 0.05), suggesting that increasing nutrient availability results in more dissimilar communities than increases in C availability.

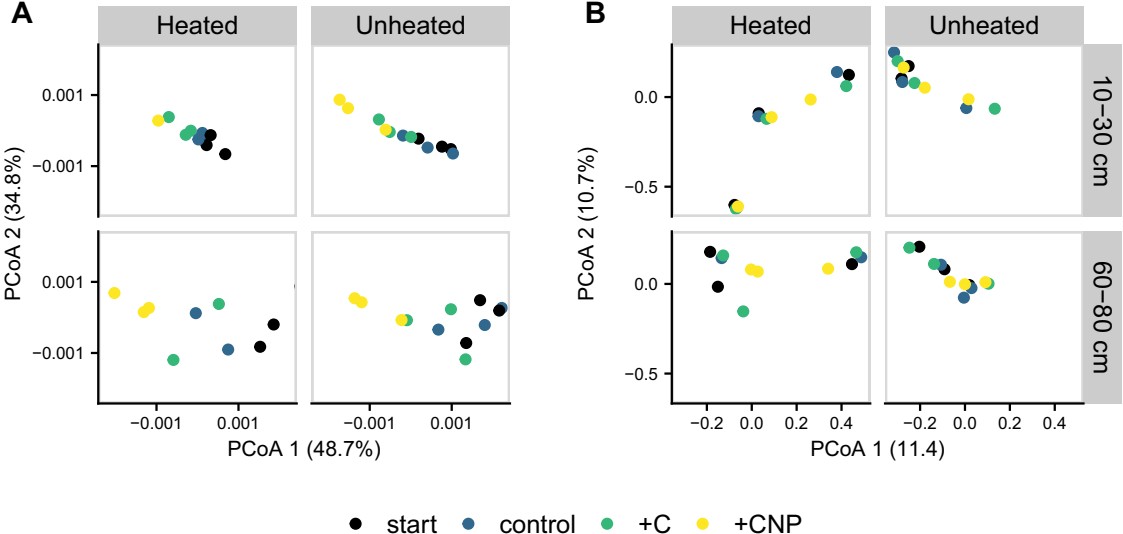

● start   ● control   ● +C   ● +CNP

**Fig. 4 Change in community composition during the 30-d incubation.** Incubations were amended with carbon (C) or carbon, nitrogen, and phosphorus (CNP) compared to the starting conditions and unamended control for prokaryote (**A**) and fungi (**B**) across the heating treatments and depths. Points denote individual samples.

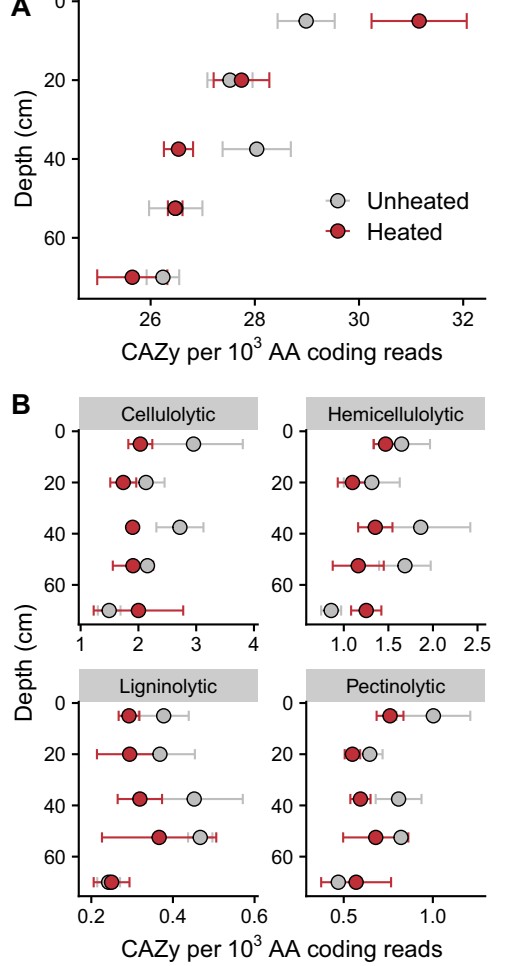

**Fig. 5 Distribution and composition of carbohydrate-active enzymes (CAZy) genes throughout heated and unheated soil profiles.** Total abundance of CAZy genes (**A**) and abundance of functionally classified CAZy genes (**B**) corrected by amino acid (AA) coding reads across heating treatments and depths. Error bars represent standard error of the mean ($n = 3$).

**Warming leads to altered substrate preferences indicating carbon limitation**. Using shotgun metagenomics, we investigated changes in functional gene composition with in situ heating throughout the soil profile. The effect of warming on the total abundance of genes encoding for carbohydrate-active enzymes (CAZy[32]) interacted with depth (MEM: $p = 0.014$; Fig. 5A). The total CAZy abundance was 7.5% higher in the heated plots only in the 0–10 cm horizon (MEM: $p < 0.001$; Fig. 5A). We classified a subset of the genes into functional groups based on previously documented C substrate[33–35] specificity of the enzymes that they code for. The abundance of genes encoding for cellulolytic enzymes was, on average, 16% lower in the heating treatment (MEM: $p = 0.032$, Fig. 5B), but was not affected by depth ($p = 0.157$). The abundance of genes encoding for hemicellulolytic, ligninolytic, and pectinolytic enzymes did not significantly differ by heating treatment or depth (MEM: $p > 0.050$; Fig. 5B). As such, the increase in total CAZy abundance in the surface soils could not be attributed to a specific substrate group.

**Microbial functional genes change with depth but not with warming**. The abundance of N-cycling genes varied with depth but, for the most part, did not change with heating (MEMs: $p > 0.050$, Supplementary Fig. 8). Generally, genes involved in oxidation reactions increased with depth (NH$_3$ oxidation MEM: $p = 0.014$, NO$_2^-$ oxidation MEM: $p < 0.001$), while genes involved in reduction reactions decreased with depth (N$_2$O reduction MEM: $p < 0.001$, NO reduction MEM: $p < 0.001$, NO$_2^-$ reduction MEM: $p = 0.036$, NO$_3^-$ reduction MEM: $p < 0.001$). This was expected because at greater depths, C concentrations decreased dramatically[3] and, in these soils, gravimetric water content decreased with depth (MEM: $p = 0.012$, Supplementary Table 3). Thus, chemoautotrophy with oxygen as the primary electron acceptor would be energetically favored. The effect of depth on the abundance of N fixation genes interacted with the heating treatment (MEM: $p = 0.033$), such that the abundance of N fixation genes was significantly higher in the heating treatment only in the surface soils (MEM: $p = 0.033$). Particulate CH$_4$ oxidation genes increased with depth (MEM: $p = 0.026$), while soluble CH$_4$ oxidation and sulfur oxidation genes did not change with depth (both MEMs: $p > 0.050$, Supplementary Fig. 8). The

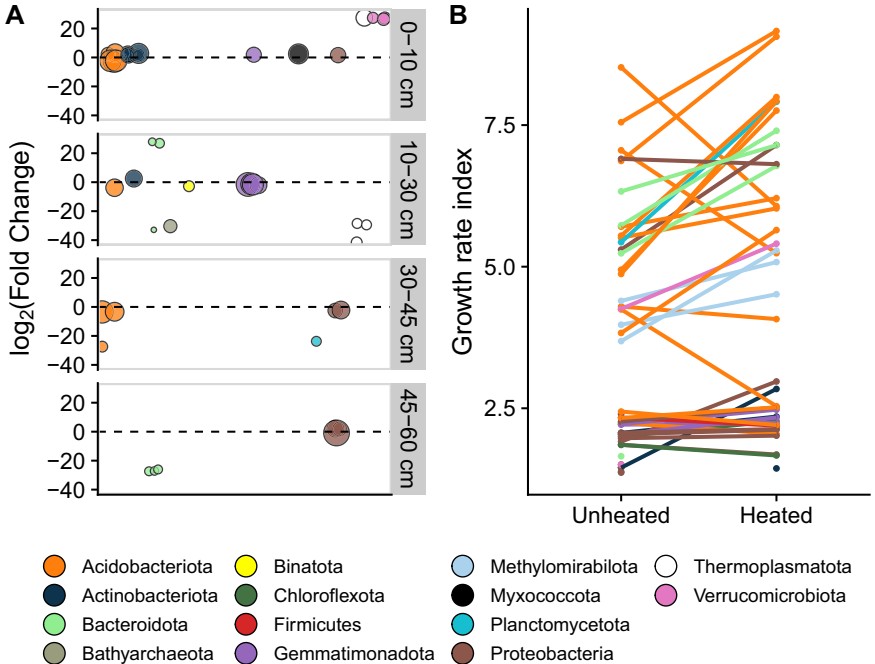

**Fig. 6 Change in abundance and growth rate of metagenome-assembled genomes (MAGs) with soil heating.** MAGs with significant changes in abundance by soil heating (Wald test: $p < 0.050$) are shown by phylum and across depths (**A**; note no significantly different MAGs were found in the 60–80 cm soil depth). Points denote individual MAGs and sizes represent the average relative abundance of the MAG in the unheated plots. The average growth rate index across all depths is plotted for each MAG shown by phylum (**B**).

abundance of the $CH_4$ and sulfur oxidation genes was unaffected by the heating treatment (MEM: $p > 0.050$).

**Higher microbial growth rates lead to greater compositional changes of MAGs in the surface compared to the subsoil.** We reconstructed 235 MAGs to determine genome-resolved responses to warming and changes in resource availability. The relative taxonomic distribution of MAGs was similar to profiles obtained via amplicon sequencing (mean relative abundance of phyla: $r = 0.79$, $p < 0.001$; Supplementary Figs. 4A, 9). The relative abundance of Dormibacteria, Firmicutes, Bathyarchaeota, and Thermoplasmatota MAGs increased with depth, while the relative abundance of Actinobacteria, Myxococcota, and Proteobacteria MAGs decreased with depth (Spearman's rho: $p < 0.050$, see Supplementary Table 4 for complete statistics, Supplementary Fig. 9). However, differences in the functional capacity of MAGs among depths were not detected. Within this diverse set of MAGs, the metabolic capacity was most frequently found as acetate fermentation and aerobic CO oxidation (Supplementary Data 1). Only 24 MAGs had complete tricarboxylic acid (TCA) cycles; however, all could utilize the metabolite 2-oxoglutarate that lies at the intersection between many C and N metabolic pathways including the TCA cycle[36]. The oxidative and non-oxidative phase of the pentose phosphate pathway were well represented in MAGs. Strikingly, 110 MAGs were capable of mineralizing both C and N from organic matter via hexosaminidases or nitrile hydratases (Supplementary Data 1). Hexosaminidase is involved in the peptidoglycan degradation, and nitrile hydratase enables utilization of nitriles as the sole source of C and N. Among oligosaccharide degradation genes, β-mannosidase was the most frequently detected (31 MAGs). β-mannosidase catalyzes the hydrolysis of β-1,4-mannosidic bonds in mannan, which is the second largest component of hemicellulose in plant cell walls[37]. Besides CO oxidation, we found that methanol and formate oxidation were key C1 metabolism

pathways, with 135 MAGs capable of either. Sulfur oxidation and sulfate reduction genes were present concomitantly in MAGs, whereas thiosulfate oxidation had a more limited representation. This set of MAGs did not contain ammonia or methane oxidizers, methanogens, or denitrifiers. Eighty-six MAGs contained ureases that facilitates urea hydrolysis to ammonia.

Warming resulted in 43 differentially abundant MAGs throughout the soil profile (Wald test: $p < 0.050$; Fig. 6A), with the number of differentially abundant MAGs decreasing with depth. Only MAGs from phyla Actinobacteria and Verrucomicrobia increased in abundance in response to warming, while remaining taxa did not show a clear trend. Across large spatial temperature gradients, genome size has been found to correlate negatively with soil temperature[38]; however, positive, negative, and neutral heat-responding MAGs had similar genome sizes (filtered by 75% completion: positive = 3.95 Mb, negative = 4.36 Mb, neutral = 3.80 Mb; ANOVA: $p = 0.515$; Supplementary Data 2). The functional potentials of energy production, C degradation strategy, and general metabolism were consistent among positive, negative, and neutral heat-responding MAGs (PerMANOVA: $p > 0.050$, Supplementary Data 3).

The responsiveness of compositional shifts among depths could be moderated by differences in microbial growth rates. To test this, we calculated estimated growth rates for each MAG across depths and heating treatments resulting in 453 growth rate estimates for 48 MAGs. Estimated growth rates decreased with depth (MEM: $p = 0.031$), such that MAGs in the 60–80 cm depth had 47% lower growth rates than the 0–10 cm depth (Supplementary Fig. 10); this result suggests that subsoil microbes were slower in their growth than were surface microbes. Growth rate increased, on average, by 16% for each MAG with soil heating (MEM: SE = 4%, $p = 0.004$, Fig. 6B). The largest increase in growth rate with heating was detected in an Acidimicrobiia MAG (phylum: Actinobacteria) originating from the 10–30 cm depth, which almost doubled in estimated growth rate (GRiD values = 1.45–2.84). This aerobic MAG contained many of the frequently

identified key C and N cycle genes in Blodgett MAGs (Supplementary Fig. 11). Besides potentially utilizing β-mannosidase, xylose, and starch as C, urea as N, and organic phosphoesters as P resources, the Acidimicrobiia MAG had CO and formate oxidation and arsenic reduction capabilities. Similar to Actinobacteria MAGs, Bacteroidota MAGs also consistently had increased growth rates with heating, on average increasing about 25% (Fig. 6B). Even though soil heating resulted in higher estimations of growth rates for individual MAGs, when considered collectively, heating did not significantly increase average estimated growth rate at each depth (Supplementary Fig. 10). Taken together with our CUE results, these findings show a modest change in growth strategies of soil microbes with multi-year warming and with short-term changes in resource availability.

## Discussion

Limited understanding of how increasing soil temperatures affect microbial metabolism, especially at depth, leads to uncertainties in determining climate–C feedbacks and predictions for soil-C sequestration[1,8]. Here, we provide a depth-informed outlook on the consequences of soil warming and associated changes in resource availability on microbial metabolism.

Surprisingly, altered resource availability of C or nutrients minimally affected microbial community composition and metabolic potential. Changes in the microbial response to soil warming over time have been attributed, in part, to selection from shifting substrate availabilities, especially those considered labile[39]. Indeed, like other studies[40,41], we found decreased potential for labile C degradation (i.e., cellulose) in the upper soil layers with soil warming, which also corroborates earlier work at this site that found enhanced decomposition and depletion of free particulate soil organic matter with warming[5]. However, response ratios of community change from our amendment-incubation experiment did not support the hypothesis that the heating treatment would interact with resource-induced changes in microbial community composition. Analysis of MAGs illustrated that microbial populations were well-adapted to degrade a wide range of complex C and N sources via hydrolytic processes throughout the soil profile (Supplementary Data 1). Such capacity may have reduced our ability to experimentally alter resource conditions by adding easily degradable cellobiose and readily assimilable ammonium and phosphate, especially under short time periods. Nevertheless, the consistent presence of genomic potential to harness complex C and N sources throughout the depth profile, and in heated soils, suggests that the direct effect of increased temperature may remain a stronger influence than warming-induced changes to resource availability.

Altering resource availability, however, affected microbial activity in ways that suggest warming of surface soils results in greater C limitation over time. Consistent with our hypothesis, the maximal respiration was more C limited in the heated plots than in the unheated plots, while the maximal respiration in the soils from the unheated plots was more nutrient limited than in those from the heated plots. Greater CAZy abundance in the warmed plots is indicative of greater C demand[42]. Furthermore, a decrease in cellulolytic CAZy genes represents a shift away from relatively labile C sources that may have been preferentially consumed during the first four years of soil warming[40,41].

The relative lack of response of N-cycling genes to the warming treatment was surprising because that treatment had higher N availability than found in unheated control soils (Supplementary Table 3), likely originating from increased decomposition of organic compounds. However, the MAGs showed that a large proportion of the community was capable of mineralizing N from organic matter, so N availability based on inorganic N may be an underestimate. Warming-altered N availability has been shown to modify N-cycling genes in other ecosystems[41,43]. However, this relative resistance of N-cycling genes might be due to relatively stronger C limitations. Heterotrophic microorganisms in Mediterranean dry forests, such as our study site, are generally considered C limited[44]. Where N is more limited, such as high-latitude forests[45], opposite trends may occur where warming-induced changes in nutrient availability may dominate the control of microbial metabolism. In the future, persistent C limitation at our site would likely result in an attenuation of elevated respiration due to warming. However, if substrate preferences shift, such C limitations could be alleviated, resulting in non-monotonic patterns of microbial respiration (e.g., Mellilo et al.[10]). Differences between the surface and subsurface respiration response may mask such patterns, underscoring the need for long-term, whole-profile soil warming studies.

Our results demonstrate that the response of the microbial community to in situ warming varies with depth in temperate soils. This is significant because previous studies of the impact of soil warming at depth are confined to tundra soils[43,46], which, overall, have higher concentrations of labile C. Therefore, the deep soils in our study represent microbial communities under a relatively large C limitation. Because of this, four and a half years of warming not only shifted microbial community composition throughout the soil profile, but responses between surface and subsoils were fundamentally different. Surface soil changes were characterized by community reorganization across multiple phyla at the OTU scale, while overall diversity remained similar between control and heated plots. We also observed a general increase in Actinobacteria and Ascomycota abundances at the phylum scale with warming. This is similar to other studies reporting that phylotypes that respond positively to heat are spread across multiple phyla[17,47], with a near global increase in Actinobacteria with warming[17,48]. However, our observed increase in the ratio of Ascomycota to Basidiomycota fungi with warming is largely uncorroborated by other studies. In fact, Basidiomycota has been found to increase in relative abundance with soil warming[49], highlighting potential differences in fungal responses among ecosystems and experiments. In contrast, warming subsoil microbial communities resulted in generally negative prokaryote differential abundance responses, suggesting that subsoil microbial communities were unable to capitalize on the new, warmed conditions. Laboratory warming (+10 °C) of Tibetan soils showed that subsoil microbial communities are, in general, less responsive to altered temperatures, at least in the short-term[50] (30 days). Our in situ results support these findings over a much longer warming period.

Differences observed in the abundance of microbial communities (at either OTU- or MAG-scale) with warming could be attributed to stochastic or neutral processes[51] that do not necessarily relate to niche-based adaptations to increased temperatures. Lack of specific genetic adaptations or phylogenetic selection in MAGs that responded positively to the heating treatment contrasts with earlier work showing that oligotrophic traits, such as low predicted rRNA operon copy number, are favored in warming experiments[17]. In the only other genome-resolved metagenomics study of experimentally warmed soil, Johnston et al.[43] showed that MAGs that increased in abundance with warming encode for both labile and complex C degradation, a result that was corroborated in warmer soils along a tundra soil temperature gradient[46]. The temperate soils in our study had lower C concentrations than the Arctic tundra, and the functional potentials of energy production and C degradation strategy were consistent among positive, negative, and neutral heat-responding MAGs. This highlights the scarcity of warming studies using

genome-resolved metagenomics, and future work is needed to determine the robustness of patterns across ecosystems, and if the functional potential of MAGs that respond positively to warming is generally unaltered over short time periods (<1 decade).

Differences in the response of surface and subsoils are likely due, in part, to the metabolic differences between microbial communities in these horizons. We found an overall increase in Acidiobacteria, Chloroflexi, and Dormibacterota (formerly Candidate Phylum AD3) relative abundance at depth, which is a finding that is consistent across numerous ecosystems[6,7,52]. Relatively slower growth rates at depth (Supplementary Fig. 10) are likely drivers of the resistance of microbial community composition to warming. Furthermore, slower growth rates were corroborated by our finding that subsoil microbial communities had relatively lower CUE, which would favor respiration over biomass growth. These results suggest that through decreased growth, subsoil microorganisms may not be able to capitalize on altered environmental conditions, slowing rates of community turnover.

Another possible reason for altered microbial community structure with warming could be the indirect effect of warming on soil moisture. At the time of sampling, gravimetric water content (kg $H_2O$/kg oven-dry soil) was about 20% lower in heated plots (unheated surface soil: $0.21 \pm 0.01$ [standard error of the mean]; heated surface soil: $0.17 \pm 0.03$; unheated subsoil: $0.16 \pm 0.01$, heated subsoil: $0.13 \pm 0.01$), which is consistent with long-term differences in volumetric water content at our site due to the heating treatment[5]. In seasonally dry ecosystems such as this Mediterranean site, soil moisture can affect microbial community structure and function throughout the soil profile[53]; however, we did not find significant correlations between soil moisture and microbial community composition. This analysis was limited to the conditions at the time of sampling and may not be indicative of longer-term soil moisture patterns. Nevertheless, the potential for moisture impacts[54] supports the need for long-term multi-factorial experiments that can elucidate interactions between soil warming and soil moisture[55].

Decreased microbial CUE at depths below 60 cm represents a unique finding because microbial CUE is rarely measured at depth. In fact, we know of only one study to have measured microbial CUE below the surface soil. In two temperate forest soils, microbial CUE (assessed by $^{18}O$ incorporation and microbial respiration) was found to be relatively constant with depth down to 40 cm[56]. However, numerous soil conditions continue to change with increasing depth, such that differences in biogeochemical conditions and microbial communities at 60 cm could represent a significant threshold for changes in microbial CUE. It is possible that lower soil moisture in subsoil horizons reduced CUE through decreased substrate diffusion or cellular dessication[54]. However, we did not detect differences in CUE between heated and unheated soils, which also differed in soil moisture. These differences in microbial physiology likely play a role in the trajectory of the microbial community in response to warming, underscoring the need for investigating the response of subsoil microbial communities to global change.

Previous studies have shown that CUE may decrease in response to warming and resource limitation[14,15]. Contrary to these previous results and our expectations, CUE was resistant to both warming and changes in resource availability in our study. Genome size has been shown to correlate negatively with CUE[57], but we did not observe any significant differences in predicted genome sizes of MAGs associated with heated or unheated soils (Supplementary Data 2). The discrepancy among studies could be due to the various methods of CUE measurement, which assess the efficiency of microbial C utilization at different metabolic scales[58]. The method applied here, which estimates the efficiency

of glycolysis and the TCA cycle through metabolic modeling, is proposed to be stable across different environmental conditions[28]. This is because this method does not include changes in C costs associated with the depolymerization of polymeric C compounds into simple sugars. Hence, changes in CUE associated with extracellular enzyme production were not well captured with our method. By incorporating our results with findings using other methods, we can partition the effects of soil heating on different aspects of microbial metabolism. This partitioning suggests that the direct and indirect effects of warming on CUE are likely derived from differences in C costs associated with extracellular enzyme production, maintenance respiration, and overflow respiration rather than changes in the efficiency of cellular respiration. This highlights the importance of using multiple measurements of CUE at different scales[58] to comprehensively understand the impact of global change on microbial metabolism.

An important caveat is that the results represent a single sampling date. Microbial communities change seasonally and may respond to treatments differently depending on climatic conditions[59]. We attempted to remedy this by sampling during a relatively wet period where microbial communities would be the most active and show the greatest response to the heating treatments. Furthermore, these results must be interpreted within the context of continued change. Nonmonotonic patterns of long-term soil respiration measurements have been observed[10], and such patterns may also occur for microbial community composition and metabolism; indeed, our results show possible mechanisms for this. These gaps underscore the need for continuous long-term measurements of the microbial response to increased temperatures.

We show that 4.5 years of $+4\,°C$ warming moderately affects soil microbial community composition and metabolism throughout the soil profile, which likely affects whole-profile C release[5]. While it remains difficult to disentangle the highly correlated direct and indirect effects of temperature on microbial communities[19], our results indicate that indirect effects, namely altered resource availability, with warming may be only partially responsible for the surface soil microbial response to warming. Microbial metabolism that allows for the decomposition of various complex soil-C stocks likely muted compositional changes in the microbial community, especially in the subsoil. Collectively, our results demonstrate a modest restructuring of the microbial community composition in surface soil accompanied by relatively little community change in the subsoil to altered soil conditions under a warming climate. We suggest that deep soils may lag in their temperature acclimation[60], potentially allowing for continued enhanced microbial respiration rates that further increase atmospheric $CO_2$ levels.

## Methods

**Study site, field experiment, and soil sampling.** The University of California Blodgett Experimental Forest is located in the foothills of the central Sierra Nevada near Georgetown, CA at 1370 m above sea level. Mean annual precipitation is 1774 mm, with most of the precipitation occurring as snow from November through April. Mean annual air temperature is about 12.5 °C[5]. The experiment is in a thinned, 80-year-old even-aged mixed conifer forest. Dominant overstory species include ponderosa pine (*Pinus ponderosa*), sugar pine (*Pinus lambertiana*), incense cedar (*Calocedrus decurrens*), white fir (*Abies concolor*), and Douglas-fir (*Pseudotsuga menziesii*). The soils are Alfisols of granitic origin with a developed O horizon as part of the Holland-Bighill complex[5].

The field experiment is explained in detail in Hicks Pries et al.[5] and in the Supplementary Methods. Briefly, the experimental design consisted of three paired heated plots and unheated control plots (circular, 3 m in diameter). The heating treatment warmed the soil 4 °C above ambient temperatures to 1 m depth while maintaining the natural temperature gradient with depth.

In June 2018, we sampled one soil core from each plot using a 5-cm-diameter AMS corer (AMS Inc. Hayward Falls, ID, USA) and extracted the following depths: 0–10 cm, 10–30 cm, 30–45 cm, 45–60 cm, and 60–80 cm (Fig. 1A). Polycarbonate

core sleeves were sterilized prior to sampling and were used only once. Samples were homogenized in sterile bags and a subsample was immediate placed on dry ice for DNA extraction. The remaining sample was placed on "blue" ice (4 °C) and refrigerated in the laboratory for 2 days until the laboratory incubation experiment.

**Soil C and N.** The remaining field-collected soils were air-dried and ground to a fine powder using mortar and pestle. Approximately 10 mg of oven-dry, ground soil was weighed into tin capsules, and these samples were analyzed for total C and N by continuous-flow, direct combustion and mass spectrometry using the ECS 4010 CHNSO analyzer (Costech Analytical Technologies, Inc., Valencia, CA, USA) at the University of California Merced Stable Isotope Laboratory (https://research.ucmerced.edu/core-facilities/stable-isotope-laboratory). Data characterizing the C and N concentrations of our samples can be found in Supplementary Table 3.

**DNA extraction.** Total DNA was extracted from frozen field-collected soils and from stored (4 °C) laboratory soils before and after the incubation (see below). To extract DNA, we used the DNeasy PowerSoil Kit (Qiagen, Germantown, MD, USA) with 0.25–0.50 g of field-moist soil (0.25 g for 0–10, 10–30, and 30–45 cm depths and 0.5 g for 45–60 and 60–80 cm depths) with minor modifications as follows. Prior to bead beating, the samples were incubated in bead-solution at 65 °C for 5 min. Cells in the samples were disrupted by bead beating with a 1600 MiniG (SPEX Sample Prep, Metuchen, NJ, USA) at a setting of 1500 rpm for 60 s, and the DNA was then further purified according to the kit protocol. DNA amounts were quantified using the Qubit dsDNA HS assay (Invitrogen, Carlsbad, CA, USA).

**Amplicon sequencing and analysis.** 16S rRNA genes were amplified in polymerase chain reactions (PCRs) using primers (F515/R806) that target the V4 region of the 16S rRNA gene, and ITS genes were amplified in PCR reactions using ITS1f/ITS2 primers[61] (Supplementary Table 5). The reverse PCR primer was barcoded with a 12-base Golay code[62]. The PCR reactions contained 2.5 µl Takara Ex Taq PCR buffer, 2 µl Takara dNTP mix, 0.7 µl Roche BSA (20 mg/ml), 0.5 µl each of the forward and reverse primers (10 µM final concentration), 0.125 µl Takara Ex Taq Hot Start DNA Polymerase (TaKaRa, Shiga, Japan), 1.0 µl genomic DNA (10 ng/reaction), and nuclease-free water in total volume of 25 µl. Reactions were held at 98 °C for 3 min. to denature the DNA, followed by amplification for 25 cycles at 98 °C for 30 s, 52 °C for 30 s, and 72 °C for 60 s; a final extension of 12 min. at 72 °C was added to ensure complete amplification.

Each sample was amplified in triplicate, combined, and purified using the Agencourt AMPure XP PCR purification system (Beckman Coulter, Brea, CA, USA). The purified amplicons were quantified using the Qubit dsDNA HS assay, and the size of the amplicons was determined using a Bioanalyzer with Agilent DNA 1000 chips (Agilent Technologies, Santa Clara, CA, USA). Amplicons were pooled (10 ng/sample) and sequenced on one lane of the Illumina Miseq platform (Illumina Inc, San Diego, CA, USA), resulting in 300 bp paired-end reads. Sequences were grouped into operational taxonomic units (OTUs) based on 97% sequence identity. For 16S rRNA gene analysis, OTUs were given taxonomic assignments in QIIME[62] version 1.7.0 using the SILVA[63] database 132. For ITS genes, representative sequences identified via UPARSE[64] (v. 11) were blasted against NCBI Nucleotide database using MegaBLAST under default parameters[65]. Resulting reports were manually curated and OTU tables were generated for further statistical analysis. Mycorrhizal taxa were assigned using FUNGuild[66] (v. 1.0).

**Metagenomic sequencing and analysis.** We generated 30 metagenomes from field-collected soils using a Kapa Biosystems LTP Library Preparation Kit for Illumina Platforms (Wilmington, DE, USA). We sheared 500 ng of genomic DNA using a Covaris S220 (Covaris, Woburn, MA, USA) with settings 140 PIP, 10.0 duty factor, 200 cycles/burst, and 65 s. Library preparation was conducted as described in the Kapa Biosystems protocol. Samples that had less than 70 ng DNA after library preparation were amplified using the reagents and recipe described in the Kapa protocol. The PCR products were purified using Agencourt AmPure XP Beads (Beckman Coulter, Indianapolis, IN, USA). Final libraries were analyzed for size using a Bioanalyzer High Sensitivity kit (Agilent, Santa Clara, CA, USA), and had a final product size ranging from 200 to 1500 bp. Libraries were pooled and sequenced using the Illumina NovaSeq paired end-read technology (at the UCSF Center for Advanced Technology, CA, USA).

Raw reads were trimmed and quality filtered using Trimmomatic[67] (v. 0.36), and read taxonomy was classified using Kaiju[68] (v. 1.6). Coverage and read diversity (herein known as Nonpareil diversity [$D_{NP}$]) of each metagenome was assessed using Nonpareil[30] (v. 3.3.3). Prodigal[69] (v. 2.6.1) was used to predict coding regions from the reads. The translated proteins from all detected coding regions of each metagenome were annotated by searching against carbohydrate-active enzymes using the CAZy database[32] (CAZyDB.07312018) via DIAMOND BLASTp[70] (v. 0.8.36) (options: -k 1 -e 1E-5-sensitive). Nitrogen cycling, methane production and oxidation, and sulfate reduction genes were annotated via hmmer[71] (v. 3.1b2) to the Kyoto Encyclopedia of Genes and Genomes (KEGG) database (downloaded 28-May-2019; https://www.kegg.jp/kegg/download/). Gene abundances (gene counts per KEGG Orthology—KO) were normalized to the

number of amino acids detected in each metagenome. On average, we obtained 8.1 Gbp (SE = 0.5 Gbp) of sequencing effort across 30 metagenomes representing three heated and three unheated soil profiles. Coverage estimated by Nonpareil[30] increased from 44.9% (SE = 1.5%) in the 0–10 cm horizon to 58.9% (SE = 2.7%) in the 60–80 cm horizon. However, coverage estimates for individual samples were relatively similar (< 2-fold differences, Supplementary Table 6), permitting comparison among samples[31].

Samples from each depth were co-assembled using MEGAHIT[72] (v. 1.1.3) with a minimum contig length of 1000 bp. Then, each individual sample was mapped back to the MEGAHIT contigs with BBmap[73] (version 36), and we extracted unmapped reads. Next, unmapped reads were concatenated and re-assembled using SPAdes[74] (v. 3.13.0) in the "meta" setting. The newly assembled contig folds were merged with the MEGAHIT contigs. Genome fragments that were larger than 1 kb were clustered into MAGs following Xue et al.[75] using MaxBin[76] (v. 2.2.5, default) and MetaBAT2[77] (v. 2.12.1, –minContig 1500), and MAGs were dereplicated using DASTool[78] (v. 1.1.10). Potential misbinnings were identified with CheckM[79] (v. 1.0.8, lineage-specific workflow), and bins were further refined to remove potentially misbinned contigs following Xue et al.[75]. FastANI (v. 0.1.2) was used to compare MAGs across assemblies and for dereplication[80].

Genome bin completeness and contamination are reported in Supplementary Data 2. We extracted nine high-quality (>90% completeness, <5% contamination) and 226 medium-quality (>50% completeness, <10% contamination) draft MAGs from our metagenomes, representing 16 bacterial and archaeal phyla[81] (Supplementary Data 2). The genome bin sizes were between 1.0 and 8.1 Mbp, with a variable GC content ranging from 38–74%. On average, 36.2% of our metagenomic reads could be mapped back to our MAGs, with MAGs constituting a larger proportion of the reads in deeper soil layers compared to surface soils (0–10 cm: 17.4%, 60–80 cm: 40.8%). Each bin was annotated with Kaiju[68] (v. 1.6.2) using default parameters utilizing the NCBI nr database and GTDB-Tk[82] (v. 0.3.2, database release 89) to classify each contig into a taxonomic rank from phylum to species. Protein-encoding genes from MAGs were predicted with Prodigal[69] (v. 2.6.1), and the resulting nucleotide sequences were searched against the KEGG database reference sequences using DIAMOND BLASTX[70] (v. 0.8.36), RAST-Tk[83] (v. 1.073 as implemented in KBase[84]) and METABOLIC[85] (v. 1.1). We estimated the growth rate of MAGs >75% completeness in each sample using GRiD (v. 1.3), discarding all values >10 and values >3 for MAGs < 4 Mbp to reduce the likelihood of reporting false growth estimates[86].

**Laboratory incubation: experimental design.** To determine the relative C and nutrient demand, we conducted a laboratory incubation experiment with the collected soils (Fig. 1B). Upon returning to the laboratory, soils from the 10–30 cm (representing "surface soils," mix of A and AB genetic horizons) and from the 60–80 cm depths (representing "subsoils", Bt genetic horizon) were split into 5-g subsamples. Each of these subsamples were then placed in 35 ml centrifuge tubes and given either a C, C+ nutrient, or control amendment in a 0.4 ml solution. These amendments represented ~10 times the microbial C and N and ~20 times the microbial P found in these soils (Supplementary Table 1). We doubled the amount of P to account for P sorption onto mineral surfaces. As such, surface aliquots received: 15 mg cellobiose-C and 4.82 mg NaCl (+C); 1.75 mg NH₄Cl-N, 0.5 mg NaH₂PO₄-P, and 15 mg cellobiose-C (+CNP); or 4.82 mg NaCl (control). Sodium chloride was given to the +C and control treatments to account for the mass of Na⁺ and Cl⁻ ions in the +CNP treatment and to create osmotically similar solutions. Subsoil subsamples were given the same amendments but at levels 23% of the surface soil, commensurate with microbial biomass differences between these two depths. We assessed nutrient limitation via the ratio of CNP- to C-amended soils rather than the ratio of nutrient-only amended soils to control soils because added C enhances nutritional demands and, thus, increases the sensitivity in detecting nutrient limitations (see Sullivan et al.[26]). Because we were interested in the relative resource demand, not absolute, we used the ratio of +CNP to +C to determine nutrient demand and +C to control to determine C demand.

The N addition in the CNP treatment in our study is much larger than many global change field experiments, which aim to mimic N deposition. However, our goal was to elucidate differences in N demand, which requires larger additions to detect[26]. Our N addition, roughly equivalent to 35 g N m⁻², falls within the range of other studies with this goal (Supplementary Table 7).

Subsamples were covered loosely with Parafilm to minimize moisture loss while allowing for gas exchange. Soils from heated and control plots were separated and placed into two different incubators (Series KB 240, Binder Group LP, Camarillo, CA, USA) set to 14 and 10 °C, respectively, corresponding to the average temperatures of the field experiment at the time of sampling.

**Laboratory incubation: microbial respiration.** We measured microbial respiration at 1, 3, 8, 16, and 28 days after the amendment addition. Four subsamples from each amendment × temperature × depth × replicate combination were uncovered and placed in a 500 ml wide-mouth screw-top jar fitted with a butyl rubber septum. We immediately sampled 16 ml of headspace and placed the jars in their respective incubators. We again sampled 16 ml of headspace after 6 h to determine the hourly respiration rate, and subsamples were removed from each jar and recovered with Parafilm. We used the same subsamples for respiration

measurements throughout the incubation. Headspace samples were then analyzed for carbon dioxide by gas chromatography using a Shimadzu GC-2014 equipped with a thermal conductivity detector (Shimadzu Corporation, Columbia, MD, USA). Cumulative gas fluxes were determined by integrating the rates at different times throughout the incubation over the course of 28 days.

**Laboratory incubation: nitrogen availability**. We determined N availability by net N mineralization during our 30-d incubation[87]. Briefly, inorganic N pool sizes before and after the 30-day incubation were determined by extracting a 5-g soil subsamples with 25 mL of 2 M KCl. Extracts were analyzed for ammonium (Lachat method: 12-107-06-1-B) and nitrate (Lachat method: 10-107-04-1-A) color-imetrically using the Lachat AE Flow Injection Auto analyzer (Lachat Instruments, Inc., Milwaukee, WI, USA). Net N mineralization rates were determined by the net changes in inorganic N pools over the 30-day incubation period[87]. Data characterizing the net N mineralization of our samples can be found in Supplementary Table 3.

**Laboratory incubation: microbial carbon use efficiency**. We used the $^{13}C$ isotopologue-metabolic modeling approach following Dijkstra et al.[28] to measure CUE at the end of the laboratory incubation. Unlike other CUE measurements that quantify the CUE at the community- or ecosystem-scale (*sensu* Geyer et al.[58]), the $^{13}C$ isotopologue-metabolic modeling approach measures the efficiency of glycolysis and the TCA cycle, independent of C substrate and extracellular C degradation. Briefly, four aliquots per replicate of each treatment were given one of four 2.5 mM, 0.5 ml amendments: 1-$^{13}C$ pyruvate; 2,3-$^{13}C$ pyruvate; 1-$^{13}C$ glucose; or U-$^{13}C$ glucose as metabolic tracers. Each aliquot was placed in a 500 ml wide-mouth Mason jar fitted with a butyl rubber septum. After 2 h for surface soils and 12 h for subsoils (because of slower metabolic rates at depth), we sampled 25 ml of headspace into an evacuated 20 ml vial. These samples were then analyzed for $^{13}CO_2$ concentrations using a G2131-i Cavity Ring Down Spectrometer $CO_2$ Isotope and Concentration Analyzer equipped with a Small Sample Isotope Module (Picarro Inc., Santa Clara, CA, USA). The ratios between $^{13}CO_2$ production rates from glucose and pyruvate isotopologues were calculated and used to model C allocated for energy production or biosynthesis.

**Statistical analysis**. All statistical analyses were conducted in R (v. 4.0.2) using the car[88], lme4[89], phyloseq[90], DESeq2[91], and vegan[92] packages. Significance was determined at the $\alpha = 0.05$ level for all statistical tests.

Biogeochemical pools and rates, response ratios, functional gene abundances (normalized by number of prodigal predicted amino acid encoding genes), and alpha diversity metrics (Chao1 and Shannon's indices for amplicons and $D_{NP}$ for metagenomes) were assessed using mixed models with the heating treatment and depth (center of horizon) as fixed effects and each paired plot as a random effect. Estimated growth rates were assessed similarly using mixed models, but with MAGs as an additional random effect. Mixed models were assessed for normality and equal variance using QQ-plots and residual vs. fitted plots, respectively. Where models did not meet these assumptions, the dependent variables were transformed, which satisfied the assumptions of normality and equal variance (e.g., N-cycling abundances were square-root transformed and net N mineralization was natural-log transformed).

Beta diversity for amplicon data was assessed by perMANOVA after proportional normalization using UniFrac distance[93] for 16S and Bray-Curtis distance for ITS. We used different distance metrics between these two marker genes because polyphyly is widespread within the fungal kingdom[94], and Bray-Curtis distances do not incorporate phylogeny (unlike Unifrac). We visualized differences in community composition using principal coordinate analysis. Correlations between microbial community composition and the edaphic variables were assessed using distance-based redundancy analysis with Bray-Curtis distances on proportionally normalized data. The estimated fold change of microbial OTU, CAZy, and MAG count abundance was assessed using Wald tests and shrinkage estimation for dispersions[91].

**Reporting summary**. Further information on research design is available in the Nature Research Reporting Summary linked to this article.

## Data availability

All sequence data are deposited at European Nucleotide Archive. 16S rRNA gene sequences can be found under the Bioproject PRJEB39495 (SAMEA7090473) and ITS gene sequences be found under the Bioproject PRJEB39496 (SAMEA7090474). The metagenome raw reads can also be found under the Bioproject PRJEB39497 (SAMEA7090475- SAMEA7090504), Metagenome-assembled-genomes and their annotations available through KBase narrative (KBase account required): https://narrative.kbase.us/narrative/57664. The following databases can be accessed using the following links: CAZy (http://www.cazy.org/) (CAZyDB.07312018), GTDB-Tk (v. 0.3.2) (https://github.com/Ecogenomics/GTDBTk), KEGG (https://www.kegg.jp/kegg/download/) (downloaded 28-May-2019), NCBI nr (https://www.ncbi.nlm.nih.gov/), SILVA (https://www.arb-silva.de/) (database 132).

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

## Acknowledgements

This work was supported as part of the Terrestrial Ecosystem Science Program by the Office of Science, Office of Biological and Environmental Research, of the U.S. Department of Energy under contract DE-AC02-05CH11231 to Lawrence Berkeley National Laboratory. N.C.D. was supported by the Department of Energy Office of Science Graduate Student Research Program, which is administered by the Oak Ridge Institute for Science and Education (DE-SC0014664), and postdoctoral development funds provided by Oak Ridge National Laboratory (Oak Ridge National Laboratory is managed by UT-Battelle, LLC, for the United States Department of Energy under contract DE-AC05-00OR22725). We thank Y. Li, J. Soong, and C. Castanha for sampling assistance and J. Jacobi-Trivia, R. Porras, and N. Canejo for laboratory experimental support. Additionally, we thank the Lawrence Berkeley National Laboratory Terrestrial Ecosystem Science Scientific Focus Area team (https://tes.lbl.gov/) as well as E. Brodie, K. Treseder, A. Asefaw Berhe, and J.M. Beaman for their insights and discussion. This manuscript has been authored by UT-Battelle, LLC under Contract No. DE-AC05-00OR22725 with the U.S. Department of Energy. The United States Government retains and the publisher, by accepting the article for publication, acknowledges that the United States Government retains a nonexclusive, paid-up, irrevocable, world-wide license to publish or reproduce the published form of this manuscript, or allow others to do so, for United States Government purposes. The Department of Energy will provide public access to these results of federally sponsored research in accordance with the DOE Public Access Plan (http://energy.gov/downloads/doe-public-access-plan).

## Author contributions

N.C.D., M.S.T., S.C.H., and N.T. designed the study. Sample collection and laboratory work was conducted by N.C.D. and N.T. N.C.D. and N.T. analyzed the data and wrote the manuscript with critical input from S.C.H. and M.S.T. All authors contributed to the article and approved the submitted version.

## Competing interests

The authors declare no competing interests.
