## [Peer Review File · Nature Communications]

Reviewer comments, first round:

Reviewer #1 (Remarks to the Author):

The authors take advantage of a long-term whole-profile soil warming experiment to investigate the impact of soil warming on community structure, physiology and metabolism of microbes in surface and deep soils. Basically, the authors took soil samples from two soil layers (one surface layer, the other sublayer), and then conducted a laboratory incubation. They looked at resource limitation, carbon use efficiency, microbial community compositions and metagenomes in different soil layers. They found a depth-informed influence of soil warming and resource availability on microbial community structure and metabolism. Overall, the manuscript was well-written, and the datasets are interesting. However, I have three major concerns about results and conclusions of the manuscript, which should be addressed before considered for publications in Nature Communications:

First, In Fig. 2A, I notice that microbial respirations of unheated treatment were much higher than that of heated treatment in the 60-80cm soil depth. This result is contradictory to many of previous findings in the literature as well as the current conclusions drawn by the authors in the manuscript. The authors did not provide a solid explanation for this, or it might be due to the unappropriated experimental design.

Second, the authors draw a conclusion that "subsoil microbial communities were even less responsive to warming", but this conclusion cannot be fully supported by current data and analyses. Current results only support subsoil microbial communities may response differently to warming, but not unresponsiveness of subsoil. In addition, results from an experiment with time series of measurements (at least 3 years) with samples along the whole soil profile must be provided to support the current conclusion.

Lastly, to make a solid conclusion claimed in the current paper, there should be a strong linkage between microbial community structure and metabolisms under climatic warming.

A few of additional comments and suggestions are provided as follows:

Line 132-149: Descriptions of statistical analyses are not clear in this paragraph. When a 'p' value is presented, the authors need to state the statistical method (e.g. ANOVA). It seems that Fig. 2B presents an overall analysis while fig. 2C presents analysis in the first two time points, but I wonder why results in the last time point are not presented here. From the experimental design, I know the replicates were not paired, but how can the error bars in Fig. 2C be obtained? In addition, I recommend the authors to conduct a multiple comparison analysis for Fig. 2B&C and mark the results in the figure.

Line 176-177: The authors used Chao1 index to tests alpha diversity, and did not find changes in the heated treatment. As far as I know, Chao1 index emphasizes on species occurrence but not species abundance. I recommend the authors to try some abundance-based alpha diversity indices (e.g. Shannon Entropy), and maybe changes can be detected. In addition, further beta diversity analyses between control and heated treatment may help to support further conclusions.

Line 194-196: The authors stated that nutrient limitation was greater than C. But the response ratio can be influence by the nutrient concentration or C addition, so I think this conclusion could not be reached.

Lines 215-218: Better to explain why genes involved in oxidation reactions increased with depth and genes involved in reduction reactions decreased with depth.

Line 304: The authors stated that "Greater CAZy abundance in the warmed plots is indicative of greater C demand". This description is contradictory to results presented in Fig. 5.

Not all microbial functional genes could be expressed normally in soil. Additional quantitative RCP (qPCR or RT-qPCR) data of the focused genes in the metagenomics data may also help consolidate the current conclusions.

Lines 650, 660, 671, 675, 702, 725, 735, 749: the formats of references are not consistent with others.

Fig. 4: Scaling in Fig.4A should be reorganized.

Fig. 5: The line diagrams can be converted to point diagrams because changes of CAZy abundance with depths are obvious not linear. In Fig.5B, the authors can add results of labile carbon if

possible. According to results in this figure, CAZy abundance in heated groups were lower (or at least not higher) than control in most cases, which is an interesting finding. I hope the authors can explain this phenomenon in discussion.

Reviewer #2 (Remarks to the Author):

General comments: The theme of this manuscript is critically important, particularly the focus on deep soils and their role in carbon cycling. A strength of the study is the use of a soil warming experiment that warms the entire soil profile to a depth of a meter or more. However, I found the manuscript lacking in a number of respects. The abstract doesn't clearly articulate the study objectives and key results. The introduction lacks sufficient context for a general audience, it skips around, and the most interesting, novel aspect (i.e., deep soils) is not the lead. That is to say, the introduction fails to provide a compelling rationale for the novelty of the work. I also have concerns about the method used to measure microbial carbon use efficiency--the approach used here has been shown to be insensitive to changing environmental conditions and is not considered the optimal method for comparing across treatments. Also, the analysis of the microbial data, particularly that for the fungal community is superficial. Focusing at the phylum level for fungi isn't particularly interesting or relevant, particularly given the focus on carbon cycling. In summary, I think there's a lot of potential in this manuscript, but it hasn't yet been realized in how the manuscript is structured.

Specific comments:

Title: The term "adaptation" should be removed since you didn't look at evolutionary change in this study.

L29. Should read "microbial metabolism and community composition". The term "microbial composition", like "plant composition" is imprecise.

L30. "even less responsive" is unclear. Relative to what?

L32-33. "muting the expected effect" is unclear. What was the expected effect and why?

L37. By "adaptation", I think you mean "acclimation". The term adaptation should be reserved for evolutionary change which you didn't evaluate in this study.

L42-44. This sentence lacks context for the uninitiated reader

L52-54. I think there's fairly good evidence that both mechanisms are at play.

L120. I suggest sticking with "microbial respiration" since the term "soil respiration" is typically used in the context of soil warming studies to refer to in situ respiration that includes both microbial and root components. Until I read the methods section, I was confused by exactly what was done in this study.

L161-162. The rationale for this work isn't well set up in the introduction.

L170-172. I find this result very surprising since it's well documented that fungal communities change significantly with depth. In particular, you should have seen a shift in the relative abundance of mycorrhizal taxa with depth, along with a shift in saprotrophic relative abundance and the saprotrophic taxa present. This would be particularly relevant to your question on C availability/quality. Did you look at fungi below the phylum level? If not, you're missing a large opportunity. Just focusing at the phylum level isn't super interesting or relevant.

L573-582. This method has been shown to give similar results regardless of the treatment (i.e., it is not very good at discerning actual differences in CUE). That is, it is insensitive to changing conditions. I would interpret these results with caution and clearly articulate the limitations of this

method.

Reviewer #3 (Remarks to the Author):

The paper by Dove et al., performed a 4.5 year-long soil warming experiment in an forest in the central Sierra Nevada, California in order to document the adaptation of the soil microbial communities to the warming, especially those of deeper soils. Overall, this reviewer believes that there is merit to this study since it is on an important topic (that of soil microbial community adaptation to climate warming), such warming experiments are challenging to perform and require substantial amount of work, and dry/Mediterranean-climate forest ecosystems have been understudied to date. However, there are alternative hypothesis (e.g., moisture effects) that are more likely to explain the results observed than the ones favored by the authors (increase efficiency of complex compounds), or the authors need to provide more unequivocal data in support of their favorable hypothesis. It would also benefit the paper to include some more direct comparisons to other deep-soil warming experiments, especially in the tundra, and comment more on how specific to the type of forest ecosystem studied the results obtained may be (or not). Finally, the text is confusing at several places and it does not using an optimal flow (but mixes topics and results often). The flow and clarity could be improved by making it clear what is the take home message at each point and linking related messages together. I dont think the taxonomic shifts with depth or warming treatment are very useful for the main thesis of the paper and thus, can be moved to the supplement, which would also shorten the paper without much loss of clarity.

Specific comments.

Ln 32-33. "muting the expected effect of altered resource availability". Not sure what this means. Perhaps rephrase? Also if the soil microbes could utilize complex organic compounds in response to warming (as mentioned later in the abstract and elsewhere), why you say "they were less responsive" to warming? Maybe, the microbes did not do the expected response but what exactly this response was?

Ln 113-117. To what extent these findings are specific to the system studied? Also, I suspect that the soil studied here is very carbon limited as a mineral soil and deep layer; hence, CUE is already very high at this depth and the limited change observed could be simply due to the carbon limitation. It will also be important to compare to tundra soils where there is a lot of carbon in the deeper layers (e.g. Johnston et al, PNAS 2019 and related studies by Virginia Rich and colleagues).

Ln 122-123. Figure 1 describes well the experiment that was performed but this information is not provided in the main text. I would suggest making the text independent of the figure.

Ln 155. Is this difference in CUE significant really? Could it be just a spurious finding? Range of values appear to be overlapping.

Ln 178-179. Did warming altered water content different with dept? Seems like an important parameters for CUE to mention, up front.

Ln 249-255 and elsewhere. Reporting of functions lacks quantitative values (e.g., how much of a difference) and statistical support (significant or not).

Ln 260. Please mention how this was performed briefly and how robust the measurement really is. Sounds like it is based on read mapping but in soil this idea does not apply well because organisms grow too slowly to show any difference between origin and end of replication (e.g., 1-2 replications per year).

Ln 262. Based on what test?

Ln 343-345. Will be nice to contrast more the deep soil studied here relative to the deeper layer of tundra, just above the permafrost. Carbon and nutrients should be much less limiting for growth in the former vs. the latter soils.

Ln 379-381. I agree that moisture may be an important factor here, likely the most important of all factors, and unfortunately, it has been under-emphasized by the authors overall, I believe.

Ln 509-511. Some of this information like metagenome sequencing effort and % reads annotated/mapped, and the Nonpareil coverage of the sampled microbial communities by sequencing, could be reported in the Results section, up front, to better set up the stage for reporting the remaining results.

REVIEWER 1

Comment 1.1: In Fig. 2A, I notice that microbial respirations of unheated treatment were much higher than that of heated treatment in the 60-80cm soil depth. This result is contradictory to many of previous findings in the literature as well as the current conclusions drawn by the authors in the manuscript. The authors did not provide a solid explanation for this, or it might be due to the unappropriated experimental design.

Response 1.1: We thank reviewer for pointing out this ambiguity. We report on (lines 147 – 148):

“Cumulative respiration was highest in the CNP-amended soils across depths and heating treatments (Mixed-effects model [MEM]: $p < 0.050$, Figure 2A).”

At the 60-80 cm depth, cumulative respiration was similar between heating treatments ($p = 0.824$). In the main text, we now include the statistics for the effect of heating in our incubation for the two depths studied. We add this text found on lines 148 through 151 below:

“Furthermore, cumulative respiration was higher in heated surface soils compared to unheated surface soils (MEM: $p < 0.001$), but cumulative respiration was similar between heating treatments in the subsoils (MEM: $p = 0.824$, Figure 2A).”

Comment 1.2: the authors draw a conclusion that “subsoil microbial communities were even less responsive to warming”, but this conclusion cannot be fully supported by current data and analyses. Current results only support subsoil microbial communities may respond differently to warming, but not unresponsiveness of subsoil. In addition, results from an experiment with time series of measurements (at least 3 years) with samples along the whole soil profile must be provided to support the current conclusion.

Response 1.2: We agree with the reviewer that a time-series approach might be more suitable in understanding the progression in microbial responses with respect to the heating treatment. As our study uses data from a single time point, microbial responses presented here represent the treatment effects up until the time of sampling. Even with their limitations (i.e., a single time point), these results show not only a different response in deep soils than surface soils but also specifically quantify under what conditions the differences were observed. For example, we found that respiration in heated soils was mostly constrained by C limitation, but this was only apparent in surface soils (10-30 cm) and not in subsoils (60-80 cm). As we agree with the reviewer, we have amended our interpretations and conclusions to demonstrate how subsoils respond *differently* than surface soils on lines 29 and 118.

Comment 1.3: To make a solid conclusion claimed in the current paper, there should be a strong linkage between microbial community structure and metabolisms under climatic warming

Response 1.3: Throughout the manuscript we provide several lines of evidence to link microbial community structure and metabolism to the direct and indirect effects of warming, one of our key findings. At multiple points throughout the manuscript, we link our metabolic findings with our community structure findings to show that the microbial response to warming is partially dependent on the metabolic response. In the Results section (lines 315 through 321), we highlight an individual MAG that had the greatest positive response to warming and briefly discuss its metabolism to speculate why it responded so positively. Reproduced below:

“The largest increase in growth rate with heating was detected in an *Acidimicrobiia* MAG (phylum: *Actinobacteria*) originating from the 10-30 cm depth, which almost doubled in estimated growth rate (GRiD values = 1.45 to 2.84). This aerobic MAG contained many of the frequently identified key C and N cycle genes in Blodgett MAGs (Figure S11). Besides potentially utilizing β -mannosidase, xylose, and starch as C, urea as N, and organic phosphoesters as P resources, the *Acidimicrobiia* MAG had CO and formate oxidation and arsenic reduction capabilities.”

In the discussion, we meld metabolic and microbial community structure results to show that the microbial response to warming is partially dependent on the metabolic response. For example, on lines 340 through 350 we state:

“However, response ratios of community change from our amendment-incubation experiment did not support the hypothesis that the heating treatment would interact with resource-induced changes in microbial community composition. Analysis of MAGs illustrated that microbial populations were well-adapted to degrade a wide range of complex C and N sources via hydrolytic processes throughout the soil profile (Table S5). Such capacity may have reduced our ability to experimentally alter resource conditions by adding easily degradable cellobiose and readily assimilable ammonium and phosphate, especially under shorter time periods. Nevertheless, the consistent presence of genomic potential to harness complex C and N sources throughout the depth profile and in heated soils suggests that the direct effect of increased temperature may remain a stronger influence than warming-induced changes to resource availability.”

Additionally, on lines 409 through 418, we state:

“Differences in the response of surface and sub- soils are likely due, in part, to the metabolic differences between microbial communities in these horizons. We found an overall increase in *Acidiobacteria*, *Chloroflexi*, and *Dormibacterota* (formerly Candidate Phylum AD3) relative abundance at depth, which is consistent across numerous ecosystems^{6,7,51}. Relatively slower growth rates at depth (Figure S10) are likely drivers of the resistance of microbial community composition to warming. Furthermore, slower growth rates were corroborated by our finding that subsoil microbial communities had relatively lower CUE, which would favor respiration over biomass growth. These results suggest that through decreased growth, subsoil microorganisms may not be able to capitalize on altered environmental conditions, slowing rates of community turnover.”

And, finally, on lines 470 through 475 we state:

“While it remains difficult to disentangle the highly correlated direct and indirect effects of temperature on microbial communities¹⁹, our results indicate that indirect effects, namely altered resource availability, with warming may be only partially responsible for the surface soil microbial response to warming. Microbial metabolism that allows for the decomposition of various complex soil C stocks likely muted compositional changes in the microbial community, especially in the subsoil.”

We reason that, collectively, these linkages provide the basis that the microbial response to warming is partially dependent on the metabolic response, a novel and important finding. Therefore, we have not changed the text significantly in response to this comment.

Comment 1.4: Line 132-149: Descriptions of statistical analyses are not clear in this paragraph. When a ‘p’ value is presented, the authors need to state the statistical method (e.g. ANOVA).

Response 1.4: We thank the reviewer for pointing this out. As this was a concern of Reviewer 3 as well (see Comment 3.12), we have added the statistical test in the parenthetical phrase used for each p-value if the name of the test did not precede the p-value in the main text. The statistical methods are explained in detail in Methods section (lines 667-688).

Comment 1.5: It seems that Fig. 2B presents an overall analysis while fig. 2C presents analysis in the first two time points, but I wonder why results in the last time point are not presented here.

Response 1.5: Here, we show the results from the peak activity (days 3 and 8) and the effect on the cumulative respiration. We focused on days 3 and 8 because as mentioned in the text (on lines 155 through 157):

“Because resource limitations may be most apparent during maximal rates of activity²⁶, we also analyzed resource limitations of respiration during days 3 and 8, when the rate of respiration was near its maximum (Figure S1).”

Figure 2B shows respiration and calculated response ratios in response to nutrient addition over the duration of the experiment (30 days), as we aimed to test the hypothesis that warming-induced soil C depletion results in increased C limitation and decreased nutrient limitation of the soil microbial community. In order to do so, we would need to consider the entire duration of the experiment not just the end point. We elaborate on the results from all measured points in the text, on lines 163 through 165 (reproduced below).

“Differences in resource limitations between heating treatments across all other date and depth combinations were not significant ($p > 0.050$), suggesting that differences in the relative resource limitations were only apparent during maximum activity and in surface soils.”

Comment 1.6: From the experimental design, I know the replicates were not paired, but how can the error bars in Fig. 2C be obtained? In addition, I recommend the authors to conduct a multiple comparison analysis for Fig. 2B&C and mark the results in the figure.

Response 1.6: The authors concede that the amendment incubation experiment was not immediately clear in the Results section (Reviewer 3 had a similar comment—see Comment 3.7). We have now added the following text on lines 129 through 134:

“Field-moist soils from each plot and horizon were split into subsamples and given either a control amendment (distilled water), a C amendment, or a C + nutrient amendment in a 0.4 ml solution in concentrations corresponding to ~10 times the microbial C and N and ~20 times the microbial P found in these soils (Table S1). Both in heated and control treatments, soil horizons were tested in triplicate for each amendment resulting in 36 incubations (2 field treatment × 2 soil horizons × 3 amendments × 3 replicates).”

Additionally, we agree with the reviewer that marking multiple comparisons in this figure would be useful, and it has now been amended.

Comment 1.7: Line 176-177: The authors used Chao1 index to tests alpha diversity, and did not find changes in the heated treatment. As far as I know, Chao1 index emphasizes on species occurrence but not species abundance. I recommend the authors to try some abundance-based alpha diversity indices (e.g. Shannon Entropy), and maybe changes can be detected. In addition, further beta diversity analyses between control and heated treatment may help to support further conclusions.

Response 1.7: We agree that using multiple measures of alpha diversity would be useful. On lines 209 through 210 and 221 through 223, we have now added the effect of heating and depth on Shannon’s diversity. We have also amended Figure S5 with this information. Overall, the impact of depth and heating on Shannon and Chao1 indices for both prokaryotes and fungi were largely the same supporting our earlier conclusions

Additionally, we have now added further beta diversity analyses by adding distance-based redundancy analyses (db-RDAs) that look at the effect of soil gravimetric water content, total carbon, and total nitrogen on microbial community structure. We found that while the overall db-RDA for 16S was significant ($p = 0.011$), none of the variables when assessed individually were ($p < 0.05$). The db-RDA for fungi was not significant ($p = 0.667$). We have added the following text to lines 181 through 186:

“While total C, total N, and gravimetric water content (GWC) together correlated with prokaryote community composition, explaining 41% of the variation (distance-based redundancy analysis [db-RDA]: $p = 0.011$), these edaphic variables did not correlate with the fungal community composition (db-RDA: $p = 0.667$). However, when assessed

individually, none of these edaphic variables correlated with prokaryote community composition (marginal db-RDA: $p < 0.05$)."

Comment 1.8: Line 194-196: The authors stated that nutrient limitation was greater than C. But the response ratio can be influenced by the nutrient concentration or C addition, so I think this conclusion could not be reached.

Response 1.8: Through the laboratory incubation we found that "that respiration in heated soils was mostly constrained by C limitation, while unheated soils were relatively more nutrient-limited." Because we calculated response ratio of CNP- to C-amended soils rather than the ratio of nutrient-only amended soils to control soils, our results show the relative nutrient limitation under surplus C. Based on the microbial community structure, we found that "increasing nutrient availability results in more dissimilar communities than increases in C availability."

We are unclear of the issue the reviewer is pointing out ("*the response ratio can be influenced by the nutrient concentration or C addition, so I think this conclusion could not be reached*") because the ratios used here are calculated based on nutrient and carbon additions. Therefore, we have not changed the text in response to this comment.

Comment 1.9: Lines 215-218: Better to explain why genes involved in oxidation reactions increased with depth and genes involved in reduction reactions decreased with depth.

Response 1.9: We agree that such information would be a useful addition to the manuscript. We now add the following text to lines 263 through 266:

"This was expected because at greater depths, C concentrations decreased dramatically³, and in our dataset, gravimetric water content decreased with depth (MEM: $p = 0.012$, Table S3). Thus, chemoautotrophy with oxygen as the primary electron acceptor would be energetically favored."

Comment 1.10: Line 304: The authors stated that "Greater CAZy abundance in the warmed plots is indicative of greater C demand". This description is contradictory to results presented in Fig. 5.

Not all microbial functional genes could be expressed normally in soil. Additional quantitative RCP (qPCR or RT-qPCR) data of the focused genes in the metagenomics data may also help consolidate the current conclusions.

Response 1.10: The authors concede that we did not adequately clarify the CAZy analysis. Our statement of "greater CAZy abundance" refers to the *total* CAZY abundance. When we classified these genes based on C substrates, only a subset of these genes was used (not all can be classified into substrate classes). We attempt to clarify this by adding the word "total" in front of "CAZy abundance" on lines 247 and 255. Furthermore, we specify that genes used in the

substrate class analysis were only a subset of all genes. The sentence starting on line 250 now reads, “We classified a *subset* of the genes into functional groups based on previously documented C substrate^{32–34} specificity...”

We do not claim to measure gene expression in this manuscript as we did not conduct RNA analyses. Perhaps the reviewer is arguing that shotgun metagenomics do not capture all genes equally and that qPCR approach is preferable or at least complementary. The authors respectfully disagree that a qPCR approach is preferable and question its value even as a complementary dataset. First of all, it is unclear as to which C genes should be assayed. Carbon degradation involves numerous genes, so deciding upon which ones is challenging. In fact, the CAZy database describes over 340,000 genes (Lombard et al. 2013 *Nucleic Acids Research*). Clearly, a qPCR approach for even a subset of these genes is intractable. Furthermore, analyses that rely on qPCR often suffer from amplification bias based on primer design, which shotgun metagenomics do not. Finally, much of the bias in shotgun metagenomics occurs from the extraction protocol (e.g., differential cell lysis between bacteria and fungi, Moran et al. 2010 *PLOS ONE*), and therefore, PCR analyses would suffer from these same levels of bias (McLaren et al. 2019 *eLife*). Hence, the authors argue that adding a qPCR approach would not strengthen this manuscript.

Comment 1.11: Lines 650, 660, 671, 675, 702, 725, 735, 749: the formats of references are not consistent with others.

Response 1.11: We thank the reviewer for pointing out these mistakes. They have been fixed.

Comment 1.12: Fig. 4: Scaling in Fig.4A should be reorganized.

Response 1.12: We thank the author for pointing out that there is only a single tick mark on the y-axis of Figure 4B. This has now been remedied. The scaling of the x-axis of Figure 4A has been expanded to not “cut off” any of the points.

Comment 1.13: Fig. 5: The line diagrams can be converted to point diagrams because changes of CAZy abundance with depths are obvious not linear. In Fig.5B, the authors can add results of labile carbon if possible. According to results in this figure, CAZy abundance in heated groups were lower (or at least not higher) than control in most cases, which is an interesting finding. I hope the authors can explain this phenomenon in discussion.

Response 1.13: These line diagrams have been amended as the reviewer requests.

Unfortunately, the authors do not have access to data on labile carbon. However, we do refer to these data in the introduction on lines 100 through 103:

“This warming-induced reduction in SOC was mainly attributed to decreased free particulate organic matter (Soong et al. *in review*), which is more directly available to

microbial decomposition²⁴, highlighting the changing resource availabilities at this site due to warming.”

We refer the reviewer to Response 1.11. The substrate class analysis uses a subset of the total CAZy genes.

REVIEWER 2

Comment 2.1: The abstract doesn't clearly articulate the study objectives and key results.

Response 2.1: We thank the reviewer for this suggestion. The abstract has been re-written to explicitly highlight the differences between the surface and subsoil. We also added quantitative values to some of our findings where appropriate.

Comment 2.2: The introduction lacks sufficient context for a general audience, it skips around, and the most interesting, novel aspect (i.e., deep soils) is not the lead. That is to say, the introduction fails to provide a compelling rationale for the novelty of the work.

Response 2.2: We have restructured the introduction by leading with how increased temperatures may affect deep soils, and this theme of deep soils was highlighted throughout the introduction. Additionally, we have worked on transitions between paragraphs to improve clarity.

Comment 2.3: I also have concerns about the method used to measure microbial carbon use efficiency--the approach used here has been shown to be insensitive to changing environmental conditions and is not considered the optimal method for comparing across treatments.

Response 2.3: Carbon use efficiency (CUE) can be measured on multiple scales, and the method used here determines the CUE of microbial physiology (i.e., the TCA cycle, Geyer et al 2016 *Biogeochemistry*). With this method, we provide information on microbial energy processes and a physiological-scale assessment of CUE. To our knowledge, there are two comparative studies showing that CUE (measured by the method used here) was stable to a glucose amendment (Geyer et al 2019 *Soil Biology & Biochemistry*) and temperature (4-20 °C) gradient (Hagerty et al. 2014 *Nature Climate Change*) in surface soils. Both of these efforts are short-term manipulations (72 h to 4 d, respectively). Here, we test for the impact of over 4 years of warming and changes in substrate stoichiometry for 30 days on CUE, factorially. We show that the CUE of the TCA cycle does not respond to long-term heating or to sporadic changes in resource amendments. This presents a substantial addition to our current knowledge on the use of available carbon to generate energy via the TCA cycle by microbial communities. Additionally, we present the novel finding that the CUE of the TCA cycle decreases with depth.

It has been documented that the method-specific differences in CUE estimates are difficult to interpret between different sites and experimental conditions (Geyer et al 2019 *Soil Biology & Biochemistry*). However, with appropriate use and identified limitations, all CUE methods could be informative.

To further clarify what the ^{13}C isotopologue CUE method measures compared to other CUE measurements, we add the following text to the Methods section of the manuscript (lines 653 through 656):

“Unlike other CUE measurements that quantify the CUE at the community- or ecosystem-scale (*sensu* Geyer et al.⁵⁶), the ^{13}C isotopologue-metabolic modeling approach measures the efficiency of glycolysis and the TCA cycle, independent of C substrate and extracellular C degradation.”

In the Discussion section (lines 443 through 458 and reproduced below), we incorporate results from this method in the context of other CUE measurements that measure CUE at the community- and ecosystem-scale (*sensu* Geyer et al 2016 *Biogeochemistry*). Triangulation of multiple measurements can thus be used to identify mechanisms of changing microbial C partitioning under future global change (i.e., increased extracellular enzyme production).

“Contrary to our expectations, CUE was resistant to both warming and changes in resource availability as others have shown that CUE may decrease in response to warming and resource limitation^{14,15}. This discrepancy among studies is likely due to the various methods of CUE measurement, which assess the efficiency of microbial C utilization at different metabolic scales⁵⁶. The method applied here, which estimates the efficiency of glycolysis and the TCA cycle through metabolic modeling, is proposed to be stable across different environmental conditions²⁷. This is because this method does not include changes in C costs associated with the depolymerization of polymeric C compounds into simple sugars. Hence, changes in CUE associated with extracellular enzyme production were not well-captured with our method. By incorporating our results with findings using other methods, we can partition the effects of soil heating on different aspects of microbial metabolism. This partitioning suggests that the direct and indirect effects of warming on CUE are likely derived from differences in C costs associated with extracellular enzyme production, maintenance respiration, and overflow respiration rather than changes in the efficiency of cellular respiration. This highlights the importance of using multiple measurements of CUE at different scales⁵⁶ to comprehensively understand the impact of global change on microbial metabolism.”

Our method measures a specific facet of CUE, and together with other data and analysis presented here, can be used to develop a fundamental understanding of microbial metabolism and C cycling under global change conditions.

Comment 2.4: Also, the analysis of the microbial data, particularly that for the fungal community is superficial. Focusing at the phylum level for fungi isn't particularly interesting or relevant, particularly given the focus on carbon cycling.

Response 2.4: We agree that further description of the fungal taxonomic and functional response to heating and depth is warranted. We have therefore added results to the main text (lines 194 through 203 and reproduced below) and two supplemental figures describing the fungal response at lower taxonomic levels and the response of ectomycorrhizal fungi, which are important for carbon and nutrient cycling in these soils.

“At the family level, we detected a significant increase in the relative abundance of the ectomycorrhizal (EM) family *Tuberaceae* throughout the soil profile (MEM: $p = 0.003$, Figure S2). However, the overall impact of heating on EM relative abundance interacted with depth (MEM: $p = 0.029$), such that heating reduced EM relative abundance (largely due to decreased *Inocybe* spp.) only in soils 0-30 cm (MEM: $p = 0.011$, Figure S3). In the subsoils, EM relative abundance was maintained in heated plots due to the significant proportions of the EM families *Cortinariaceae* and *Rhizopogonaceae*. Hence, while the relative abundance of EM fungi did not decrease with depth (MEM: $p = 0.711$, Figure S3), the community composition of EM fungi differed with soil depth (PerMANOVA: $p < 0.001$, $R^2 = 0.052$, Figure S2).”

Comment 2.5: Title: The term "adaptation" should be removed since you didn't look at evolutionary change in this study.

Response 2.5: This is a fair point by the reviewer. The title now reads, “Phyla-wide metabolic capabilities mute positive response to direct and indirect impacts of warming throughout the soil profile”

Comment 2.6: L29. Should read "microbial metabolism and community composition". The term "microbial composition", like "plant composition" is imprecise.

Response 2.6: This phrase (on line 28) has been amended to “microbial community composition and metabolism.”

Comment 2.7: L30. "even less responsive" is unclear. Relative to what?

Response 2.7: The sentence starting on line 29 now reads, “However, the subsoil microbial communities responded differently to warming than in the surface.”

Comment 2.8: L32-33. "muting the expected effect" is unclear. What was the expected effect and why?

Response 2.8: The expected effect is that heat-induced changing resource availabilities partially explains shifts in microbial community composition in response to warming. However, given the word constraints of the abstract (150 words), we are unable to clarify this sentence. While this is

discussed in greater detail in the introduction, we agree that presenting an “expected” effect in the abstract without the proper context is inappropriate. Therefore, we have omitted the word “expected” in this sentence. It now reads (on lines 30 through 32):

“Throughout the soil profile—but notably in the subsoil— physiologic and genomic measurements showed that phylogenetically different microbes could utilize complex organic compounds dampening the effect of altered resource availability induced by warming.”

Comment 2.9: L37. By "adaptation", I think you mean "acclimation". The term adaptation should be reserved for evolutionary change which you didn't evaluate in this study.

Response 2.9: This word has been removed.

Comment 2.10: L42-44. This sentence lacks context for the uninitiated reader

Response 2.10: We agree that this sentence may not be appropriate for a broad audience. We have thus amended it to read, “For example, while short-term measurements indicate that increased temperatures enhance microbial respiration, long-term (>10 y) *in situ* measurements of microbial respiration are more nuanced^{9,10}.”

Comment 2.11: L52-54. I think there's fairly good evidence that both mechanisms are at play.

Response 2.11: We agree with the reviewer that the literature shows that both are at play. Originally, the sentence suggests that either one occurs. Therefore, we have slightly modified the sentence to clarify that the extent of each mechanisms is what is unclear. The modified sentence, on lines 64 through 67 is reproduced below:

“Indeed, long-term soil warming has been shown to alter surface soil microbial community structure and metabolism^{17,18}, but it is unclear to what extent this is caused by increased temperatures or in combination with altered availability of resources for microbial growth such as organic substrates and nutrients¹⁹.”

Comment 2.12: L120. I suggest sticking with "microbial respiration" since the term "soil respiration" is typically used in the context of soil warming studies to refer to *in situ* respiration that includes both microbial and root components. Until I read the methods section, I was confused by exactly what was done in this study.

Response 2.12: The reviewer makes a good point here. The subheading now reads, “Resource availability is a key factor affecting surface soil microbial respiration under warming.”

Comment 2.13: L161-162. The rationale for this work isn't well set up in the introduction.

Response 2.13: This comment refers to the third subheading in the results section: “The impact of warming on microbial taxonomic composition is depth dependent and resistant to shifts in resource availability.”

The introduction has been rewritten, so perhaps the improved writing will clarify our justification for the work described in this subheading. The justification for studying the direct (enzyme and growth kinetics) and indirect (i.e., nutrient-mediated) impacts of soil warming on microbial community composition is now on lines 38 through 80:

“The impact of increasing global temperatures on soil microbial communities and carbon dioxide (CO₂) emissions is still largely uncertain¹, and this uncertainty is exacerbated in the subsoil. Deep soil microbial communities are vastly understudied compared to their surface counterparts². Most soil warming studies focus on the upper soil layers where microbial activity and carbon (C) concentrations are significantly higher^{3,4}. However, the microbial response to soil warming at depth is non-negligible. For instance, over half of extracellular enzyme activity in the upper meter occurs below 20 cm⁴. Furthermore, Hicks Pries et al.⁵ recently showed that the apparent Q₁₀ (i.e., temperature sensitivity) of the microbial community was similar throughout the soil profile. When warmed, subsoil respiration accounted for 40% of the increase in CO₂ emissions from the whole soil profile⁵. Additionally, microbial community structure and composition in the subsoil are different than those in the surface soils, as subsoil microbial communities are dominated by slow-growing oligotrophs^{6,7}. Under a warming climate, uncertainties surrounding the trajectory of subsoil microbial community reorganization limit our predictive capability of future microbial states and functionalities.

The indirect effects of increased temperatures on soil microbial communities and microbial respiration further complicates our understanding of climate-carbon cycle feedbacks, making it difficult to constrain long-term models of soil C cycling⁸. For example, while short-term measurements indicate that increased temperatures enhance microbial respiration, long-term (>10 y) *in situ* measurements of microbial respiration are more nuanced^{9,10}. Besides the direct effects of increased temperature, *in situ* soil warming can also decrease C availability^{11,12}, alter C chemical composition¹³, and increase nutrient availability¹¹ in surface soils. These impacts are likely to affect rates of respiration through indirect effects on the soil microbial community. For instance, decreased C quantity and quality (i.e., number of enzymatic steps necessary to depolymerize C compounds) alters microbial community composition and decreases respiration^{14,15}. Additionally, increased nutrient availability in warmed soils decreases extracellular enzyme production for nutrient-acquiring enzymes¹⁶, which may increase biomass growth and also alter microbial community structure. Indeed, long-term soil warming has been shown to alter surface soil microbial community structure and metabolism^{17,18}, but it is unclear to what extent this is caused by increased temperatures or in combination with altered availability of resources for microbial growth such as organic substrates and nutrients¹⁹. Understanding changes to microbial metabolism in response to soil warming

may, in part, resolve some of these uncertainties, constrain model predictions⁸, and explain the disparate findings of long-term empirical observations^{1,9,10}.

Warming-induced changes to resource availability—due to increased decomposition and altered plant growth—could have an exacerbated effect on the subsoil microbial community where resource demand is the strongest. Resource availability at depth differs from surface soil, where C and nitrogen (N) availabilities decrease with depth³ because of smaller pool sizes, increased mineral protection, and increased spatial discontinuity of organic matter²⁰. However, laboratory rates of C and N mineralization of added substrates were as fast in the subsoils as in surface soils in an old-growth forest, suggesting that microbial competition and demand for C and N resources does not decrease with depth²¹. Hence, subsoil microbial communities may have as strong of substrate demands as in the surface, but mineralization in subsoil is more substrate-limited. Because over half of soil organic C (SOC) is found below a 20-cm depth³, incorporating subsoil responses to warming is critical in constraining predictions for long-term soil C storage.”

Comment 2.14: L170-172. I find this result very surprising since it's well documented that fungal communities change significantly with depth. In particular, you should have seen a shift in the relative abundance of mycorrhizal taxa with depth, along with a shift in saprotrophic relative abundance and the saprotrophic taxa present. This would be particularly relevant to your question on C availability/quality. Did you look at fungi below the phylum level? If not, you're missing a large opportunity. Just focusing at the phylum level isn't super interesting or relevant.

Response 2.14: See Response 2.4.

Comment 2.15: L573-582. This method has been shown to give similar results regardless of the treatment (i.e., it is not very good at discerning actual differences in CUE). That it, it is insensitive to changing conditions. I would interpret these results with caution and clearly articulate the limitations of this method.

Response 2.15: See Response 2.3. To further clarify what the ¹³C isotopologue CUE method measures compared to other CUE measurements, we add the following text to the Methods section of the manuscript (lines 653 through 656):

“Unlike other CUE measurements that quantify the CUE at the community- or ecosystem-scale (*sensu* Geyer et al.⁵⁶), the ¹³C isotopologue-metabolic modeling approach measures the efficiency of glycolysis and the TCA cycle, independent of C substrate and extracellular C degradation.”

REVIEWER 3

Comment 3.1: there are alternative hypothesis (e.g., moisture effects) that are more likely to explain the results observed than the ones favored by the authors (increase efficiency of complex compounds), or the authors need to provide more unequivocal data in support of their favorable

hypothesis.

Response 3.1: Given that soil moisture decreased in our heating treatments and that soil moisture has been shown to correlate with microbial community composition (Kaisermann et al. 2015 *Applied Soil Ecology*), we agree that soil moisture could be an important factor here. We interrogated the role of soil moisture on microbial community composition using distance-based redundancy analyses (db-RDAs), and we found little to no correlation of the microbial community with soil moisture. Lines 181 through 186:

“While total C, total N, and gravimetric water content (GWC) together correlated with prokaryote community composition, explaining 41% of the variation (distance-based redundancy analysis [db-RDA]: $p = 0.011$), these edaphic variables did not correlate with the fungal community composition (db-RDA: $p = 0.667$). However, when assessed individually, none of these edaphic variables correlated with prokaryote community composition (marginal db-RDA: $p < 0.05$).”

We integrated this finding into our Discussion using the original paragraph discussing the moisture effects on soil microbial community structure. However, we moved this paragraph from the end of the Discussion to the middle of the Discussion to highlight its importance. The rewritten paragraph, on lines 419 through 430, is reproduced below:

“Another possible reason for altered microbial community structure with warming could be the indirect effect of warming on soil moisture. At the time of sampling, gravimetric water content (kg H₂O/kg oven-dry soil) was about 20% lower in heated plots (unheated surface soil: 0.21 ± 0.01 [standard error of the mean]; heated surface soil: 0.17 ± 0.03 ; unheated subsoil: 0.16 ± 0.01 , heated subsoil: 0.13 ± 0.01), which is consistent with long-term differences in volumetric water content at our site⁵. In seasonally dry ecosystems such as this Mediterranean site, soil moisture can affect microbial community structure and function throughout the soil profile⁵²; however, we did not find significant correlations between soil moisture and microbial community composition. This analysis was limited to the conditions at the time of sampling and may not be indicative of longer-term soil moisture patterns. Nevertheless, the potential for moisture impacts⁵³ supports the need for long-term multi-factorial experiments that can elucidate interactions between soil warming and soil moisture⁵⁴.”

Additionally, we discussed other results from this study in the context of changing moisture to further highlight its potential importance. On lines 263 through 266 we use our soil moisture results to partially explain differences in redox processes with depth:

“This was expected because at greater depths, C concentrations decreased dramatically³, and in our dataset, gravimetric water content decreased with depth (MEM: $p = 0.012$, Table S3). Thus, chemoautotrophy with oxygen as the primary electron acceptor would be energetically favored.”

And, on lines 437 through 440, we discuss how lower soil moisture at depth could reduce CUE:

“It is possible that lower soil moisture in subsoil horizons reduced CUE through decreased substrate diffusion or cellular desiccation⁵⁵. However, we did not detect differences in CUE between heating treatments, which also differed in soil moisture.”

Overall, these results indicate that soil moisture may play a modest role in affecting microbial function, but the impact of soil moisture on microbial community composition in our study was likely minimal. However, we do concede that we did not alter soil moisture independently of soil warming, so these conclusions are based solely on correlations. We discuss the limitations of our study with regard to soil moisture on lines 426 through 430:

“This analysis was limited to the conditions at the time of sampling and may not be indicative of longer-term soil moisture patterns. Nevertheless, the potential for moisture impacts⁵³ supports the need for long-term multi-factorial experiments that can elucidate interactions between soil warming and soil moisture⁵⁴.”

Comment 3.2: It would also benefit the paper to include some more direct comparisons to other deep-soil warming experiments, especially in the tundra, and comment more on how specific to the type of forest ecosystem studied the results obtained may be (or not).

Response 3.2: Here, and in Comment 3.6, the reviewer suggests that the comparisons to multiple ecosystems should be further discussed, and we whole-heartedly agree. Throughout the discussion we now mention how edaphic properties, namely C and N availabilities, among different ecosystems may affect our findings. When discussing the impact of C and nutrient availability on microbial metabolism, we now mention (on lines 364 through 368):

“However, this relative resistance of N cycling genes might be due to relatively stronger C limitations. Heterotrophic microorganisms in Mediterranean dry forests, such as our study site, are generally considered C-limited⁴³. Where N is more limited, such as high-latitude forests⁴⁴, opposite trends may occur where warming-induced changes in nutrient availability may dominate the control of microbial metabolism.”

We also point out that previous studies of the impact of soil warming at depth are confined to tundra soils, which have relatively higher labile C availabilities. Thus, our findings could be due to higher C limitations, especially at depth. We reproduce this text (on lines 373 through 379) below:

“Our results are the first to demonstrate that the response of the microbial community to *in situ* warming varies with depth in temperate soils. This is significant because previous studies of the impact of soil warming at depth are confined to tundra soils^{42,45}, which, overall, have higher concentrations of labile C. Therefore, the deep soils in our study represent microbial communities under a relatively large C limitation. Because of this, four and a half years of warming not only shifted microbial community composition throughout the soil profile, but responses between surface and subsoils were fundamentally different.”

Additionally, in contextualizing some of our MAG results, we further compare the metabolic capabilities of positive heat-responding MAGs in our study with those from experimental (Johnston et al. 2019 *PNAS*) and observational (Woodcroft et al. 2018 *Nature*) tundra warming studies, and discuss how differences in C availability between these ecosystems may have influenced our results. On lines 399 through 408, we now state:

“In the only other genome-resolved metagenomics study of experimentally warmed soil, Johnston et al.⁴² showed that MAGs that increased in abundance with warming encode for both labile and complex C degradation, a result that was corroborated in warmer soils along a tundra soil temperature gradient⁴⁵. The temperate soils in our study had lower C concentrations than the Arctic tundra, and the functional potentials of energy production and C degradation strategy were consistent among positive, negative, and neutral heat-responding MAGs. This highlights the scarcity of warming studies using genome-resolved metagenomics, and future work is needed to determine if the functional potential of MAGs that respond positively to warming is generally unaltered over short-time periods (< 1 decade) and the robustness of patterns across ecosystems.”

Finally, we would like to highlight areas in the manuscript where we have already compared our results with others. First, on lines 337 through 340:

“Indeed, like other studies^{39,40}, we found decreased potential for labile C degradation (i.e., cellulose) in the upper soil layers with soil warming, which also corroborates earlier work at this site that found enhanced decomposition and depletion of free particulate soil organic matter with warming⁵.”

On lines 381 through 388:

“We also observed a general increase in *Actinobacteria* and *Ascomycota* abundances at the phylum scale with warming. This is similar to other studies reporting that phylotypes that respond positively to heat are spread across multiple phyla^{17,46} and a near global increase in *Actinobacteria* with warming^{17,47}. However, our observed increase in the ratio of *Ascomycota* to *Basidiomycota* fungi with warming is largely uncorroborated by other studies. In fact, *Basidiomycota* has been found to increase in relative abundance with soil warming⁴⁸, highlighting potential differences in fungal responses among ecosystems and experiments.”

On lines 390 through 393:

“Laboratory warming (+10 °C) of Tibetan soils showed that subsoil microbial communities are, in general, less responsive to altered temperatures, at least in the short-term⁴⁹ (30 days). Our *in situ* results support these findings over a much longer warming period.”

On lines 396 through 399:

“Lack of specific genetic adaptations or phylogenetic selection in MAGs that responded positively to the heating treatment contrasts with earlier work showing that oligotrophic traits, such as low predicted rRNA operon copy number, are favored in warming experiments¹⁷.”

On lines 410 through 412:

“We found an overall increase in *Acidiobacteria*, *Chloroflexi*, and *Dormibacterota* (formerly Candidate Phylum AD3) relative abundance at depth, which is consistent across numerous ecosystems^{6,7,51}.”

Overall, we assert that, with the additional discussion of patterns across ecosystems (namely those of tundra soils), we have provided sufficient context to the reader to evaluate the robustness of our results without overburdening the reader with superfluous discussion.

Comment 3.3: I also have concerns about the method used to measure microbial carbon use efficiency--the approach used here has been shown to be insensitive to changing environmental conditions and is not considered the optimal method for comparing across treatments.

Response 3.3: See Response 2.3.

Comment 3.4: I dont think the taxonomic shifts with depth or warming treatment are very useful for the main thesis of the paper and thus, can be moved to the supplement, which would also shorten the paper without much loss of clarity.

Response 3.4: Determining taxonomy-function relationships is a key goal of microbial ecology (Strickland et al. 2009 *Ecology*, Fierer et al. 2012 *PNAS*, Treseder and Lennon 2015 *Microbiology & Molecular Biology Reviews*), and thus, the reporting of both shifts in microbial taxonomy and function is warranted in the main text. Throughout the text we use shifts in taxonomy to inform changes in microbial functions (see examples in Response 1.3). Furthermore, Reviewers 1 and 2 requested further taxonomic descriptions and analyses (see Comments 1.3, 1.7, 2.4, and 2.14). Therefore, we have not changed the text in response to this comment.

Comment 3.5: Ln 32-33. “muting the expected effect of altered resource availability”. Not sure what this means. Perhaps rephrase? Also if the soil microbes could utilize complex organic compounds in response to warming (as mentioned later in the abstract and elsewhere), why you say "they were less responsive" to warming? Maybe, the microbes did not do the expected response but what exactly this response was?

Response 3.5: See Response 2.8.

Comment 3.6: Ln 113-117. To what extent these findings are specific to the system studied? Also, I suspect that the soil studied here is very carbon limited as a mineral soil and deep layer; hence, CUE is already very high at this depth and the limited change observed could be simply due to the carbon limitation. It will also be important to compare to tundra soils where there is a lot of carbon in the deeper layers (e.g. Johnston et al, PNAS 2019 and related studies by Virginia Rich and colleagues).

Response 3.6: See Response 3.2.

Comment 3.7: Ln 122-123. Figure 1 describes well the experiment that was performed but this information is not provided in the main text. I would suggest making the text independent of the figure.

Response 3.7: See Response 1.6. We have now added the following text on lines 129 through 134:

“Field-moist soils from each plot and horizon were split into subsamples and given either a control amendment (distilled water), a C amendment, or a C + nutrient amendment in a 0.4 ml solution in concentrations corresponding to ~10 times the microbial C and N and ~20 times the microbial P found in these soils (Table S1). Both in heated and control treatments, soil horizons were tested in triplicate for each amendment resulting in 36 incubations (2 field treatment × 2 soil horizons × 3 amendments × 3 replicates).”

Comment 3.8: Ln 155. Is this difference in CUE significant really? Could it be just a spurious finding? Range of values appear to be overlapping.

Response 3.8: We agree that the range of values overlaps. However, based on the distribution of topsoil and subsoil CUE, the mixed effects model rejects the null hypothesis that these distributions are statistically similar ($p < 0.001$). Although, we do concede that this is a single measurement and that the community will likely continue to transition with further warming. We think that this is an important point that warrants further discussion and can be applied to all of our measurements. Therefore, we now add the following paragraph beginning on line 459:

“An important consideration of these conclusions is that these results were obtained during a single sampling date. Microbial communities change seasonally and may respond to treatments differently depending on climatic conditions⁵⁷. We attempted to remedy this by sampling during a relatively wet period where microbial communities would be the most active and show the greatest response to the heating treatments. Furthermore, these results must be interpreted within the context of continued change. Non-monotonic patterns of long-term soil respiration measurements have been observed¹⁰, and such patterns may also occur for microbial community composition and metabolism; indeed our results show possible mechanisms for this. These gaps underscore the need for continuous long-term measurements of the microbial response to increased temperatures.”

Comment 3.9: Ln 178-179. Did warming altered water content different with dept? Seems like an important parameters for CUE to mention, up front.

Response 3.9: Water content did change with depth, and this is now mentioned on lines 264 through 265 (see Response 1.9). We further discuss how this might affect CUE on lines 437 through 440 and reproduced below in the Discussion section. Because this text refers to the interpretation of data instead of the presentation of it, we reason that it fits better within the Discussion section.

“It is possible that lower soil moisture in subsoil horizons reduced CUE through decreased substrate diffusion or cellular desiccation⁵³. However, we did not detect differences in CUE between heating treatments, which also differed in soil moisture.”

Comment 3.10: Ln 249-255 and elsewhere. Reporting of functions lacks quantitative values (e.g., how much of a difference) and statistical support (significant or not).

Response 3.10: It is unclear to the authors what the reviewer is referring to here. Lines 249 through 255 (in the original document) refer to differentially abundant MAGs for which the statistical support was given (line 251 in the original document).

It is possible that the reviewer may be referring to the functional attributes of the MAGs (lines 233 through 249 in the original document). To this end, we added further quantitative information on, and the section now reads (on lines 284 through 298):

“Only 24 MAGs had complete tricarboxylic acid (TCA) cycles; however, all could utilize the metabolite 2-oxoglutarate that lies at the intersection between many C and N metabolic pathways including the TCA cycle³⁵. The oxidative and non-oxidative phase of the pentose phosphate pathway were well represented in MAGs. Strikingly, 110 MAGs were capable of mineralizing both C and N from organic matter via hexosaminidases or nitrile hydratases (Table S5). Hexosaminidase is involved in the peptidoglycan degradation, and nitrile hydratase enables utilization of nitriles as the sole source of C and N. Among oligosaccharide degradation genes, β -mannosidase was the most frequently detected (31 MAGs). β -mannosidase catalyzes the hydrolysis of β -1,4-mannosidic bonds in mannan, which is the second largest component of hemicellulose in plant cell walls³⁶. Besides CO oxidation, we found that methanol and formate oxidation were key C1 metabolism pathways, with 135 MAGs capable of either. Sulfur oxidation and sulfate reduction genes were present concomitantly in MAGs, whereas thiosulfate oxidation had a more limited representation. This set of MAGs did not contain ammonia or methane oxidizers, methanogens, or denitrifiers. Eighty-six MAGs contained ureases that facilitates urea hydrolysis to ammonia.”

We did not conduct statistical testing for this section. Rather, this section was used to describe the metabolism of the entire MAG set (i.e., in this section we did not test for statistical differences among groups).

Later in this subsection we do test for differences in functional potential among MAGs that were classified as positive, negative, and neutral heat-responders, and the statistical support is given (line 305 through 308):

“The functional potentials of energy production, C degradation strategy, and general metabolism were consistent among positive, negative, and neutral heat-responding MAGs (PerMANOVA: $p > 0.050$, Table S7).”

Comment 3.11 Ln 260. Please mention how this was performed briefly and how robust the measurement really is. Sounds like it is based on read mapping but in soil this idea does not apply well because organisms grow too slowly to show any difference between origin and end of replication (e.g., 1-2 replications per year).

Response 3.11 While the method applied here is a relatively new, it has been empirically tested with large data sets (Emiola and Oh 2018 *Nature Communications*) and benchmarked to growth rates obtained from isolates. In practice, it will remain difficult to pinpoint the bottleneck of growth rate estimations on a natural community because we do not typically know the actual growth rates of all members of any given community. Especially for slow growing organisms, we are unlikely to overcome these bottlenecks easily. There are any number of reasons why an organism may not reproduce at its physiological maximal rate. Yet, precisely for this reason, growth rate inference from MAGs has utility as it enables us to contrast estimations from different states. Therefore, we have not changed the text in response to this comment.

Comment 3.12: Ln 262. Based on what test?

Response 3.12: We have added the test to the p-value parenthetical statements throughout the manuscript. See Response 1.4.

Comment 3.13: Ln 343-345. Will be nice to contrast more the deep soil studied here relative to the deeper layer of tundra, just above the permafrost. Carbon and nutrients should be much less limiting for growth in the former vs. the latter soils.

Response 3.13: We agree that we could further compare and contrast the results among biomes, especially tundra soils where some work has been done (other biomes are not represented in genome-centric studies of soil warming). Therefore, we have amended this area of the Discussion on lines 399 through 405 and reproduced below:

“In the only other genome-resolved metagenomics study of experimentally warmed soil, Johnston et al.⁴² showed that MAGs that increased in abundance with warming encode

for both labile and complex C degradation, a result that was corroborated in warmer soils along a tundra soil temperature gradient⁴⁵. The temperate soils in our study had lower C concentrations than the Arctic tundra, and the functional potentials of energy production and C degradation strategy were consistent among positive, negative, and neutral heat-responding MAGs.”

Comment 3.14: Ln 379-381. I agree that moisture may be an important factor here, likely the most important of all factors, and unfortunately, it has been under-emphasized by the authors overall, I believe.

Response 3.14: See Response 3.1.

Comment 3.15: Ln 509-511. Some of this information like metagenome sequencing effort and % reads annotated/mapped, and the Nonpareil coverage of the sampled microbial communities by sequencing, could be reported in the Results section, up front, to better set up the stage for reporting the remaining results.

Response 3.15: We experimented moving this sequencing information into the Results section. However, doing so broke up the continuity of the writing, which could cause confusion, diminishing the major findings of the paper. Instead, we now present some of the Nonpareil results, namely the diversity estimation with the other measures of alpha diversity on lines 206 through 210 (and reproduced below). We hope that this satisfies the reviewer’s criticism.

“We also used a read-based alpha diversity metric (Nonpareil, D_{NP} ²⁹) to compare diversity estimates between amplicon- and metagenome-based sequencing efforts. D_{NP} for all soil depths was 20.7 ± 0.6 (standard error of the mean, $n = 30$); falling within the proposed spectrum for soil metagenomes³⁰. The D_{NP} was unaltered by soil heating (MEM: $p = 0.904$, Table S2).”

Reviewer comments, second round:

Reviewer #1 (Remarks to the Author):

I am happy to see that the manuscript has been improved compared to the previous version. However, several major concerns raised in the previous reviews have not been fully addressed. First, I noticed that the authors provided a simple statistic result for the respiration data, but as mentioned previously, these datasets are perplexing without further arguments and supporting datasets. Second, the author didn't provide additional evidence to address the Comments 1.2 and 1.3, which are critical to draw a solid conclusion as the authors made in their text. For instances, as we suggested previously, many known and important functional genes could be checked using qPCR to compare with the results from metagenomic analyses. Finally, I wonder whether the authors could provide an alternative approach, or at least offering additional datasets, to the methodological problem, as it raised by the other two reviewers as well, that the CUE is not sensitive.

In sum, the authors could not draw a solid conclusion without providing strong arguments and additional supporting datasets. The manuscript must be improved before it can be accepted by Nature Communications.

Reviewer #2 (Remarks to the Author):

The manuscript is much improved and my previous comments have largely been addressed. The study addresses a critically important and understudied question--the degree to which subsoils respond to warming and underlying microbial mechanisms influencing that response. Although I still take issue with the choice of method for measuring CUE, since the method employed is the least sensitive to altered soil conditions (shown in numerous studies, including those cited by the authors), the CUE data are only a part of the overall study and the authors do a reasonable job of discussing potential methodological differences.

A few minor points:

L157-158. I would say "SOM" not "C" chemical composition (i.e., C is C). Also include soil moisture here as an indirect driver

L183. I don't think the term "prokaryote" is used any longer or at least as I understand it, is not preferred by those studying bacteria/archaea.

L187. Remove "the" before "fungal community..."

L206. I'll need to double check the methods, but if you only sequenced the ITS (which I think is what was done), then you wouldn't find AMF because the ITS locus is biased against this group. AMF need to be evaluated using 18S. I suggest changing the wording here, so you're not implying that AMF weren't present, but only that you didn't detect them because of the method used.

Reviewer #3 (Remarks to the Author):

The revised paper by Dove et al., represents a substantially improved version and the authors have addressed most of my concerns (as well as those of the other reviewers, I believe). I have a few remaining concerns however.

The response of the authors to my major comment that perhaps lower moisture, caused by warming, could account, at least in part, for the reduced Carbon Utilization Efficiency (CUE) observed as opposed to the alternative explanations favored by the authors is not very clear to me. The authors wrote:

Given that soil moisture decreased in our heating treatments and that soil moisture has been shown to correlate with microbial community composition (Kaisermann et al. 2015 Applied Soil Ecology), we agree that soil moisture could be an important factor here. We interrogated the role of soil moisture on microbial community composition using distance-based redundancy analyses (db-RDAs), and we found little to no correlation of the microbial community with soil moisture.

>Why the moisture should affect community CUE through changes in microbial community composition especially in slow growing subsoil communities that likely grow at 1 generation per year -or less- and thus, 4-5 years might not be even enough time for some changes to be detectable at the DNA level (composition)? It could affect CEU directly via desiccation or extra energy needed to absorb organic compounds under drier conditions, I think. So, the response here is not clear to me, and I did not necessarily expect changes in community composition from moisture for this specific site/soil.

And, on lines 437 through 440, we discuss how lower soil moisture at depth could reduce CUE: "It is possible that lower soil moisture in subsoil horizons reduced CUE through decreased substrate diffusion or cellular desiccation. However, we did not detect differences in CUE between heating treatments, which also differed in soil moisture."

>I thought there is only one heating treatment, that of 40C, so it is not clear to me what treatments (plural) the authors refer to. If they refer to the lab experiment to measure CUE under nutrient supplemented vs. non-nutrients, I am wondering if this is really relevant for in-situ. If they mean control vs. warming, this had did had an effect on CEU, right?

Please note that the Comment 3.3. in the rebuttal is not my comment but that of another reviewer. Apparently, the authors have mixed the reviews here. My comment was instead: 'the text is confusing at several places and it does not using an optimal flow (but mixes topics and results often). The flow and clarity could be improved by making it clear what is the take home message at each point and linking related messages together. "

>That said, the authors have made a good effort to restructure the paper and the text reads much clearer to me, and with a more logical flow.

REVIEWER #1:

Comment R1-1.1: I noticed that the authors provided a simple statistic result for the respiration data, but as mentioned previously, these datasets are perplexing without further arguments and supporting datasets.

Response R1-1.1: Statistical methods used to analyze the respiration data (Mixed-effects model [MEM]) is appropriate given our experimental design and available measurement points. We are unclear what is referred as the “*simple statistic*.” However, in addressing this comment, the authors will assume the reviewer is referring to the resource limitation statistics because these were remarked as “unclear” by this reviewer in the previous draft of the manuscript. In an attempt to address this comment, we now clarify that we are referring to “proximate” resource limitations (as opposed to “ultimate” resource limitations), and provide a supporting citation on lines 137-144. This method for estimating the proximate resource limitation is widely implemented. While citation constraints prevent us from citing the numerous papers demonstrating this, we do cite a review by Sullivan et al. (2014 *Ecology*). We hope that we have clarified our resource limitation statistics sufficiently.

Comment R1-1.2: The author didn’t provide additional evidence to address the Comments 1.2 and 1.3, which are critical to draw a solid conclusion as the authors made in their text. For instances, as we suggested previously, many known and important functional genes could be checked using qPCR to compare with the results from metagenomic analyses.

Response R1-1.2: We conceded that using the wording “less responsive” was imprecise (Comment 1.2, original responses), and we changed the wording to “responded differently”—our discussions and conclusions are also adjusted accordingly. We highlighted these in our first rebuttal. This change in wording was adjusted verbatim to what was suggested by the reviewer: “[c]urrent results only support subsoil microbial communities may **respond differently** to warming...” Hence, if the reviewer is unsatisfied with the current wording, their meaning remains unclear in Comment 1.2 (original responses from the first review round).

As for additional evidence to support Comment 1.3, we will first reproduce this comment made by Reviewer #1: “*To make a solid conclusion claimed in the current paper, there should be a strong linkage between microbial community structure and metabolisms under climatic warming.*”

As stated in our previous responses, linkages between different measurements are provided throughout the paper. What is stated by the reviewer would be an assumption or hypothesis to be tested “... *there should be a strong linkage between ...* ,” which we already included in the manuscript. At multiple points in the manuscript, we link the microbial metabolism with community structure to show that the microbial response to warming is partially dependent on the metabolic response. We point out that it remains difficult to disentangle the highly correlated direct and indirect effects of temperature on microbial communities. One of our key results that indirect effects, namely altered resource availability, with warming may be only partially

responsible for the surface soil microbial response to warming is particularly important. We reproduced all edited text for the reviewer's appraisal in our earlier rebuttal.

Furthermore, the reviewer's earlier comment on the need for qPCR was: "*Not all microbial functional genes could be expressed normally in soil. Additional quantitative PCR (qPCR or RT-qPCR) data of the focused genes in the metagenomics data may also help consolidate the current conclusions.*" We stated, in the original response letter, that we are not aiming to resolve microbial responses via RNA analysis. qPCR and RT-qPCR are not equivalent methods; both addresses completely different components of microbial functionality: earlier for the genomic potential and the later for the utilization of the potential at a given time and condition.

We further disagree with the justification previously used by the reviewer for implementing qPCR analysis. The reviewer writes, "*...many known and important functional genes could be checked using qPCR to compare with the results from metagenomic analyses.*" The earlier applications of metagenomics repeatedly confirmed that with appropriate reading depth relative abundances calculated from metagenomes are well in agreement with individual checks with qPCR. For example, a recent example would be the comparison for mercury methylation (*HgcA*) gene across different soils by Christensen et al. (2019). They use metagenomics and qPCR and state that "*Overall, assessing the Hg-methylating community by identifying hgcAB from metagenomic or PCR-based methods showed highly similar results.*" Clearly, a qPCR approach for even a subset of the genes assessed here would be intractable, and its necessity is opposed due to the large number of studies that reported similar statements to Christensen et al. (2019), the availability of primers characterizing the myriad of genes involved in complex C degradation, and primer bias (our metagenomics approach actually alleviates these latter two concerns). It is well covered in the literature that, if accuracy of microbial/gene identification is a concern, long-read sequencing is needed to overcome the issue not qPCR. We do not have any concerns on the accuracy of the gene identification methods used in this manuscript as we use conservative cut offs to minimize computational errors. Results are readily consolidated as we partition gene distributions and relative abundances of C- and N-cycle genes between different depths and treatment (unheated vs. heated), provide appropriate statistics, and discuss our results.

Comment R1-1.3: I wonder whether the authors could provide an alternative approach, or at least offering additional datasets, to the methodological problem, as it raised by the other two reviewers as well, that the CUE is not sensitive.

Response R1-1.3: We would like to reiterate that all CUE measurements vary in their ability to assess different cellular processes. The method used here determines the CUE of microbial physiology—more specifically, the TCA cycle. As discussed at length in Geyer et al. (2016) and in our manuscript on lines 452-463 and lines 658-661, we provide information on microbial energy processes and a physiological-scale assessment of CUE. We show that the CUE of the TCA cycle does not respond to chronic heating or to acute changes in resource amendments. This presents a substantial addition to our current knowledge on the use of available carbon to generate energy via the TCA cycle by microbial communities. It has been documented that the method-specific differences in CUE estimates are difficult to interpret between different sites and

experimental conditions (Geyer et al. 2019), but we argue that it is precisely these methodological differences that allow illumination of new knowledge on microbial metabolism in response to warming (lines 458-463). Furthermore, we do show significant differences in CUE with depth, suggesting that this method is not “insensitive” to differences in biological and chemical soil properties. We respectfully disagree that our CUE method is problematic (i.e., a “*methodological problem*”) and maintain our previous argument that with appropriate use and identified limitations all CUE methods can be informative.

In the revised manuscript we present an additional line of evidence supporting our finding that CUE can be resistant to warming. Others have shown that genome size is negatively correlated with CUE (Saiffudin et al. 2019). Thus, if CUE is consistent between the heated and unheated treatment, one would expect MAGs associated with each treatment to have similar genome sizes, which is what we see. On line 304, we state:

“Across large spatial temperature gradients, genome size has been found to correlate negatively with soil temperature³⁸; however, positive, negative, and neutral heat-responding MAGs had similar genome sizes (filtered by 75% completion: positive = 3.95 Mb, negative = 4.36 Mb, neutral = 3.80 Mb; ANOVA: $p = 0.515$; Table S6).”

We therefore relate the lack of differences in genome sizes between the MAGs associated with the heated vs. unheated treatments to lack of differences in CUE with the heated vs. unheated treatments on line 448:

“Genome size has been shown to correlate negatively with CUE⁵⁷, but we did not observe any significant differences in predicted genome sizes of MAGs associated with heated or unheated soils (Table S6).”

We would also like to point out that the other two reviewers appear to be satisfied with our rebuttal and discussion of the methodological differences among different CUE estimation techniques (See Comment R1-2.1). We hope that this clarification along with this new line of evidence has sufficiently addressed the reviewer’s concern about this measurement.

REVIEWER #2:

Comment R1-2.1: The manuscript is much improved and my previous comments have largely been addressed. The study addresses a critically important and understudied question--the degree to which subsoils respond to warming and underlying microbial mechanisms influencing that response. Although I still take issue with the choice of method for measuring CUE, since the method employed is the least sensitive to altered soil conditions (shown in numerous studies, including those cited by the authors), the CUE data are only a part of the overall study and the authors do a reasonable job of discussing potential methodological differences.

Response R1-2.1: The authors appreciate the feedback here, and we are happy that the reviewer recognizes our discussion of the potential methodological differences among CUE measurements.

Comment R1-2.2: L57-58. I would say "SOM" not "C" chemical composition (i.e., C is C). Also include soil moisture here as an indirect driver

Response R1-2.2: This has been amended.

Comment R1-2.3: L183. I don't think the term "prokaryote" is used any longer or at least as I understand it, is not preferred by those studying bacteria/archaea.

Response R1-2.3: It has been communicated by Associate Editor that the term "prokaryote" is appropriate. Therefore, the text has not been altered in response to this comment.

Comment R1-2.4: L187. Remove "the" before "fungal community..."

Response R1-2.4: This has been amended.

Comment R1-2.5: L206. I'll need to double check the methods, but if you only sequenced the ITS (which I think is what was done), then you wouldn't find AMF because the ITS locus is biased against this group. AMF need to be evaluated using 18S. I suggest changing the wording here, so you're not implying that AMF weren't present, but only that you didn't detect them because of the method used.

Response R1-2.5: This sentence has altered and now reads (on line 204), "Sequencing the ITS region, we did not detect arbuscular mycorrhizal fungi in our soils."

REVIEWER #3:

Comment R1-3.1: The response of the authors to my major comment that perhaps lower moisture, caused by warming, could account, at least in part, for the reduced Carbon Utilization Efficiency (CUE) observed as opposed to the alternative explanations favored by the authors is not very clear to me. The authors wrote:

"Given that soil moisture decreased in our heating treatments and that soil moisture has been shown to correlate with microbial community composition (Kaisermann et al. 2015 Applied Soil Ecology), we agree that soil moisture could be an important factor here. We interrogated the role of soil moisture on microbial community composition using distance-based redundancy analyses (db-RDAs), and we found little to no correlation of the microbial community with soil moisture."

Why the moisture should affect community CUE through changes in microbial community composition especially in slow growing subsoil communities that likely grow at 1 generation per

year -or less- and thus, 4-5 years might not be even enough time for some changes to be detectable at the DNA level (composition)? It could affect CEU directly via desiccation or extra energy needed to absorb organic compounds under drier conditions, I think. So, the response here is not clear to me, and I did not necessarily expect changes in community composition from moisture for this specific site/soil.

“And, on lines 437 through 440, we discuss how lower soil moisture at depth could reduce CUE: “It is possible that lower soil moisture in subsoil horizons reduced CUE through decreased substrate diffusion or cellular desiccation. However, we did not detect differences in CUE between heating treatments, which also differed in soil moisture.”

I thought there is only one heating treatment, that of 40C, so it is not clear to me what treatments (plural) the authors refer to. If they refer to the lab experiment to measure CUE under nutrient supplemented vs. non-nutrients, I am wondering if this is really relevant for in-situ. If they mean control vs. warming, this had did had an effect on CEU, right?

Response R1-3.1: It is possible that the authors were confused by Comment 3.1 (original responses), as Response 3.1 (original response) was in reference to changes in microbial community composition—not CUE.

Our working hypothesis was that warming resulted in changes to all soil depths. Indeed, 4.5 years was enough time to detect a moderate but significant shift in the prokaryote and fungal community composition, and shifts were different in the surface and subsoil. However, we did not find evidence for this shift being related to soil moisture.

We agree with the reviewer that the direct impacts of lower soil moisture on CUE (such as desiccation and lower substrate diffusion) could be at play here. As stated by the reviewer, we mention this in the discussion (lines 440 through 443). However, we concede that the wording is unclear. There was only one *in situ* treatment, and that was the soil warming—“treatments” referred to +4C heating and the unheated control. We have now corrected our statement on line 442, “However, we did not detect differences in CUE between heated and unheated soils, which also differed in soil moisture.”

Comment R1-3.2: Please note that the Comment 3.3. in the rebuttal is not my comment but that of another reviewer. Apparently, the authors have mixed the reviews here. My comment was instead:

“the text is confusing at several places and it does not using an optimal flow (but mixes topics and results often). The flow and clarity could be improved by making it clear what is the take home message at each point and linking related messages together.”

That said, the authors have made a good effort to restructure the paper and the text reads much clearer to me, and with a more logical flow.

Response R1-3.2: We apologize for the confusion here, but we are glad that we were able to adequately respond to your intended comment.

LITERATURE CITED

- Christensen, G. A. *et al.* Determining the Reliability of Measuring Mercury Cycling Gene Abundance with Correlations with Mercury and Methylmercury Concentrations. *Environ. Sci. Technol.* **53**, 8649–8663 (2019).
- Geyer, K. M., Dijkstra, P., Sinsabaugh, R. & Frey, S. D. Clarifying the interpretation of carbon use efficiency in soil through methods comparison. *Soil Biol. Biochem.* **128**, 79-88 (2018)
- Geyer, K. M., Kyker-Snowman, E., Grandy, A. S. & Frey, S. D. Microbial carbon use efficiency: accounting for population, community, and ecosystem-scale controls over the fate of metabolized organic matter. *Biogeochemistry* **127**, 173–188 (2016).
- Saifuddin, M., Bhatnagar, J. M., Segrè, D. & Finzi, A. C. Microbial carbon use efficiency predicted from genome-scale metabolic models. *Nat Commun* **10**, 1–10 (2019).